# EMPIRICAL ANALYSIS OF MODEL SELECTION FOR HETEROGENEOUS CAUSAL EFFECT ESTIMATION

**Divyat Mahajan**[1]        **Ioannis Mitliagkas**[1]        **Brady Neal**[*,1]        **Vasilis Syrgkanis**[*,2]

[1] Mila, Université de Montréal                    [2] Stanford University

## ABSTRACT

We study the problem of model selection in causal inference, specifically for conditional average treatment effect (CATE) estimation. Unlike machine learning, there is no perfect analogue of cross-validation for model selection as we do not observe the counterfactual potential outcomes. Towards this, a variety of surrogate metrics have been proposed for CATE model selection that use only observed data. However, we do not have a good understanding regarding their effectiveness due to limited comparisons in prior studies. We conduct an extensive empirical analysis to benchmark the surrogate model selection metrics introduced in the literature, as well as the novel ones introduced in this work. We ensure a fair comparison by tuning the hyperparameters associated with these metrics via AutoML, and provide more detailed trends by incorporating realistic datasets via generative modeling. Our analysis suggests novel model selection strategies based on careful hyperparameter selection of CATE estimators and causal ensembling.

## 1 INTRODUCTION

Several decision-making tasks require us to compute the personalized effect of interventions on an individual. If interventions are assigned based on the average effect, then it might result in sub-optimal outcomes (Segal et al., 2012) as the heterogeneity of the data is not taken into account. Hence, identifying which individuals benefit the most from an intervention would result in better policies. The emphasis on individual treatments effects has been shown in multiple domains, from personalised healthcare (Foster et al., 2011) to social sciences (Xie et al., 2012).

This has led to several techniques that estimate flexible and accurate models of heterogeneous treatment effects. These approaches range from adapting neural networks (Shi et al., 2019) to random forests (Wager & Athey, 2018), along with frameworks like double machine learning (Chernozhukov et al., 2016; Foster & Syrgkanis, 2019; Nie & Wager, 2021), instrumental variables (Hartford et al., 2017), meta learners (Künzel et al., 2019), etc. But how do we select between the different estimators? While in some situations we can choose between the estimators based on domain knowledge and application requirements, it is desirable to have a model-free approach for model selection. Further, the commonly used practice of cross-validation in supervised learning problems (Bengio et al., 2013) cannot be used for model selection in causal inference, as we never observe both of the potential outcomes for an individual (fundamental problem of causal inference (Holland, 1986)).

Towards this, surrogate metrics have been proposed that perform model selection using only observational data. Earlier proposals were based on evaluating the nuisance models associated with the estimators, and the utility of decision policy (Zhao et al., 2017) based on the heterogeneous treatment effects of the estimator. Recently, the focus has shifted towards designing surrogate metrics that approximate the true effect and compute its deviation from the estimator's treatment effect (Nie & Wager, 2021; Saito & Yasui, 2020), and they have also been shown to be more effective than other metrics (Schuler et al., 2018; Alaa & Van Der Schaar, 2019). However, most of these evaluation studies have been performed only on a few synthetic datasets, therefore, the trend in such studies could be questionable. Also, there is often a lack of fair comparison between the various metrics as

---

*Equal Advising. Correspondence to: `divyat.mahajan@mila.quebec`

some of them are excluded from the baselines when authors evaluate their proposed metrics. Hence, we have a poor understanding of which surrogate criteria should be used for model selection.

**Contributions.** In this work, we perform a comprehensive empirical study [1] over **78 datasets** to understand the efficacy of **34 surrogate metrics** for conditional average treatment effect (CATE) model selection, where the model selection task is made challenging by training a large number of estimators (**415 CATE estimators**) for each dataset. Our evaluation framework encourages unbiased evaluation of surrogate metrics by proper tuning of their nuisance models using AutoML (Wang et al., 2021), which were chosen in a limited manner even in recent benchmarking studies (Curth & van der Schaar, 2023). We also provide a novel two-level model selection strategy based on careful hyperparameter selection for each class of meta-estimators, and causal ensembling which improves the performance of several surrogate metrics significantly.

To ensure we have reliable conclusions, unlike prior works, we also make use of recent advances in generative modeling for causal inference (Neal et al., 2020) to include realistic benchmarks in our analysis. Further, we introduce several new surrogate metrics inspired by other related strands of the literature such as TMLE, policy learning, calibration, and uplift modeling.

Our analysis shows that metrics that incorporate doubly robust aspects significantly dominate the rest across all datasets. Interestingly, we also find that plug-in metrics based on T-Learner are never dominated by other metrics across all datasets, which suggests the impact of tuning the nuisance models properly with AutoML for CATE model selection.

**Notations:** Capital letter denote random variables ($X$) and small case letters ($x$) denote their realizations. The nuisance models of the CATE estimators have upward hat $\hat{\eta} = (\hat{\mu}, \hat{\pi})$, while the nuisance models of surrogate metrics have downward hat $\breve{\eta} = (\breve{\mu}, \breve{\pi})$. Potential outcomes are denoted as $(Y(0), Y(1))$ while the pseudo-outcomes are represented as $Y(\eta) = Y_1(\eta) - Y_0(\eta)$.

## 2 CATE MODEL SELECTION: SETUP & BACKGROUND

We work with the potential outcomes framework (Rubin, 2005) and have samples of random variables $(Y, W, X)$, where $X$ are the pre-treatment covariates, $W$ is the treatment assignment, and $Y$ is the outcome of interest. We consider binary treatments $W \in \{0, 1\}$, and have two potential outcomes $(Y(0), Y(1))$ corresponding to the interventions $(do(W = 0), do(W = 1))$. The observational data $\{x, w, y\}$ are sampled from an unknown joint distribution $P_\theta(X, W, Y(0), Y(1))$.

Typical causal inference queries require information about the propensity (treatment assignment) distribution ($\pi_w(x) = \mathbb{P}[W = w | X = x]$) and the expected potential outcomes ($\mu_w(x) = \mathbb{E}[Y(w) | X = x]$), commonly referred as the nuisance parameters $\eta = (\mu_0, \mu_1, \pi)$.

Our target of inference is the conditional average treatment effect (CATE), that represents the average effect of intervention $(Y(1) - Y(0))$ on the population with covariates $X = x$.

$$\text{CATE:} \quad \tau(x) = \mathbb{E}[Y(1) - Y(0) | X = x] = \mu_1(x) - \mu_0(x).$$

Under the standard assumptions of ignorability (Peters et al., 2017) the expected outcomes are identified using observational data as $E[Y(w)|X = x] = E[Y|W = w, X = x]$, which further implies CATE is identified as follows (more details in Appendix A.1):

$$\tau(x) = \mathbb{E}[Y|W = 1, X = x] - \mathbb{E}[Y|W = 0, X = x]$$

**Meta-Learners for CATE Estimation.** We consider the meta-learner framework (Künzel et al., 2019) that relies on estimates of nuisance parameters ($\hat{\eta}$) to predict CATE. E.g., if we can reliably estimate the potential outcomes ($\mathbb{E}[Y|W = w, X = x]$) from observational data by learning regression functions $\hat{\mu}_w$ that predict the outcomes $y$ from the covariates $x$ for treatment groups $w \in \{0, 1\}$, then we can estimate the CATE as follows, also known as the **T-Learner**

$$\hat{\tau}_T(x) = \hat{\mu}_1(x) - \hat{\mu}_0(x) \tag{1}$$

---

[1]The code repository can be accessed here: github.com/divyat09/cate-estimator-selection

Similarly, we could also learn a single regression function ($\hat{\mu}_{x,w}$) to estimate the potential outcomes, also known as the **S-Learner**

$$\hat{\tau}_S(x) = \hat{\mu}(x, 1) - \hat{\mu}(x, 0) \tag{2}$$

Following Curth & Van der Schaar (2021), such estimating strategies are called indirect meta-learners, as their main learning objective is to estimate potential outcomes and not CATE directly. In contrast, with direct meta-learners we learn additional regression models ($f$) to estimate CATE from covariates $X$, which provides additional regularization. One popular direct meta-learner is the **Doubly Robust (DR) Learner** (Kennedy, 2020), where we first estimate the DR pseudo-outcomes $y^{\mathrm{DR}}(\hat{\eta})$ and then learn the CATE predictor $\hat{f}$ by regressing the pseudo-outcomes on the covariates.

$$y^{\mathrm{DR}}(\hat{\eta}) = y_1^{\mathrm{DR}}(\hat{\eta}) - y_0^{\mathrm{DR}}(\hat{\eta}) \quad \text{where} \quad y_w^{\mathrm{DR}}(\hat{\eta}) = \hat{\mu}(x, w) + \frac{y - \hat{\mu}(x, w)}{\hat{\pi}_w(x)} \tag{3}$$

$$\hat{\tau}_{\mathrm{DR}} := \hat{f}_{\mathrm{DR}} = \arg\min_{f \in F} \sum_{\{x,w,y\}} \left( y^{\mathrm{DR}}(\hat{\eta}) - f(x) \right)^2 \tag{4}$$

Please refer to Appendix A for a detailed recap on meta-learners used in this study.

**CATE Model Selection.** Given a set of CATE estimates $\{\hat{\tau}_1, .., \hat{\tau}_M\}$ from estimators $\{E_1, .., E_M\}$, CATE model selection refers to finding the best estimator, $E_{m^*}$ s.t. $m^* = \arg\min_i L(\hat{\tau}_i)$, where the $L(\hat{\tau})$ denotes the precision of heterogeneous effects (PEHE) (Hill, 2011).

$$L(\hat{\tau}) = \mathbb{E}_X[(\hat{\tau}(X) - \tau(X))^2] \tag{5}$$

If we had access to counterfactual data (observed both $Y(0), Y(1)$), then we could compute the true effect $\tau(X)$ and use the ideal metric PEHE for model selection. Hence, the main difficulty stems from not observing both potential outcomes for each sample, and we need to design surrogate metrics ($M(\hat{\tau})$) that use only observational data for model selection.

**Surrogate Metrics for CATE Model Selection.** A common approach for designing surrogate metrics ($M(\hat{\tau})$) is to learn an approximation of the ground truth CATE as $\tilde{\tau}(x)$ and then compute the PEHE as follows (Schuler et al., 2018).

$$M(\hat{\tau}) = \frac{1}{N} \sum_{i=1}^{N} (\hat{\tau}(x_i) - \tilde{\tau}(x_i))^2 \tag{6}$$

Different choices for $\tilde{\tau}$ would give rise to different surrogate metrics. We briefly describe a few techniques commonly used for estimating $\tilde{\tau}$, with a more detailed description of the various surrogate metrics considered in our work can be found in Appendix B.

One class of surrogate metrics are the *plug-in* surrogate metrics which estimate $\tilde{\tau}$ by training another CATE estimator on the **validation set**, and we could employ similar estimation strategies as meta-learners. E.g., analogous to T-Learner, we can learn $\tilde{\tau}(x)$ as the difference in the estimated potential outcomes $\check{\mu}_1(x) - \check{\mu}_0(x)$, known as the T-Score (Alaa & Van Der Schaar, 2019).

$$M^T(\hat{\tau}) := \frac{1}{N} \sum_{i=1}^{N} (\hat{\tau}(x_i) - \tilde{\tau}_T(x_i))^2 \qquad \tilde{\tau}_T(x) := \check{\mu}_1(x) - \check{\mu}_0(x) \qquad (\text{T Score})$$

Another class of surrogate metrics are the *pseudo-outcome* surrogate metrics that estimate $\tilde{\tau}$ as pseudo-outcomes ($Y(\check{\eta})$). E.g., we can construct the pseudo-outcome metric using DR pseudo-outcome (E.q. 3), known as the DR Score (Saito & Yasui, 2020).

$$M^{\mathrm{DR}}(\hat{\tau}) := \frac{1}{N} \sum_{i=1}^{N} (\hat{\tau}(x_i) - \tilde{\tau}_{\mathrm{DR}}(x_i))^2 \qquad \tilde{\tau}_{\mathrm{DR}} := y^{\mathrm{DR}}(\check{\eta}) \qquad (\text{DR Score})$$

Note that DR-Learner would require training the CATE predictor ($f_{\mathrm{DR}}$) as well, however with the pseudo-outcome metrics we don't train such direct predictors of CATE. Infact, training a direct CATE predictor ($\check{f}_{\mathrm{DR}}$) for the metric as well would make it a plug-in surrogate metric.

**Which surrogate criteria to use?** While there have been several surrogate metrics proposed in the literature that enable CATE model selection using only observed data, we have a poor understanding of their relative advantages/disadvantages. Towards this, there have been a couple of benchmarking studies, the first one by Schuler et al. (2018) where they found the *R-Score* (Nie & Wager, 2021) strategy to be the most effective. However, since their work, there have been new surrogate criteria proposed (Alaa & Van Der Schaar, 2019; Saito & Yasui, 2020) and also they experimented with only a single synthetic data generation process. The latest study by Curth & van der Schaar (2023) considers an exhaustive set of surrogate metrics and analyzes their performance on a carefully designed synthetic data generation process. Their analysis shows the limitations of the factual prediction criteria that rely on evaluating the generalization of nuisance models for model selection. They also find that pseudo-outcome variants are less susceptible to congeniality bias as compared to their plug-in counterparts. In this work, we build further in the same spirit of conducting a more thorough analysis to obtain insightful trends regarding the performance of different surrogate metrics. The next section provides details on the proposed evaluation framework and highlights the associated important design choices ignored in the prior works.

## 3 FRAMEWORK FOR COMPARING MODEL SELECTION STRATEGIES

Consider a set of trained CATE estimators ($E$) and a set of surrogate metrics ($\{M(\hat{\tau})\}$), our task is to determine the effectiveness of each metric. Let $E_M^*$ denote the set of estimators that are optimal w.r.t the metric $M$, i.e., $E_M^* = \arg\min_{e \in E} M(\hat{\tau}_e)$. Then, similar to prior works, we judge the performance of any surrogate metric $M$ by computing the ideal PEHE metric (E.q. 5) for the best estimators selected by it, i.e., PEHE($E_M^*$)= $\frac{1}{|E_M^*|} * \sum_{e \in E_M^*} L(\hat{\tau}_e)$. Since PEHE($E_M^*$) determines the quality of the best estimators selected by a metric $M$, hence it can be used to compare the different surrogate metrics. We now state the novel aspects of our evaluation framework (Figure 1) for comparing the surrogate metrics for CATE model selection.

**Well-tuned surrogate metrics via AutoML.** Since surrogate metrics involve approximating the ground-truth CATE ($\tilde{\tau}$) (E.q. 6), we need to infer the associated nuisance models ($\check{\eta}$) on the validation set. The nuisance models ($\check{\eta}$) play a critical role in the performance of these metrics as sub-optimal choices for them can lead to a biased approximation ($\tilde{\tau}$) of the true CATE. Despite its importance, tuning of metric's nuisance models is done by searching over a small manually specified grid of hyperparameters in prior works. Hence, we use AutoML, specifically FLAML (Wang et al., 2021) to select the best-performing nuisance model class as well as its hyperparameters. Since AutoML can select much better nuisance models than grid search or random search would for the same amount of compute, the surrogate metrics would have less tendency to be biased.

**Two-level model selection strategy.** The set of trained CATE estimators can be grouped based on the different learning criteria. E.g., consider the population of CATE estimators to be comprised of two groups, where the first group $E_T = \{\hat{\tau}_T(\hat{\eta}_1), \cdots, \hat{\tau}_T(\hat{\eta}_m)\}$ contains all the estimators of type T-Learner and the second group $E_{DR} = \{\hat{\tau}_{DR}(\hat{\eta}_1), \cdots, \hat{\tau}_{DR}(\hat{\eta}_n)\}$ contain all the estimators of type DR-Learner. Given a surrogate metric $M(\hat{\tau})$, prior works select over the entire estimator population, $E_M^* = \arg\min_{e \in E_T \cup E_{DR}} M(\hat{\tau}_e)$, which we term as *single-level model selection strategy*.

However, another approach would be to first select amongst the estimators within each meta-learner using a criterion *better suited for that specific meta-learner*, and then select over the remaining population of meta-learners using the surrogate metric. In the example above, we could use $T$ Score to select amongst the T-Learner group, i.e., $E_T^* = \arg\min_{e \in E_T} M^T(\hat{\tau}_e)$. Similarly, we could use DR Score to select amongst the DR-Learner group, i.e, $E_{DR}^* = \arg\min_{e \in E_{DR}} M^{DR}(\hat{\tau}_e)$. Then we could select between $E' = E_T^* \cup E_{DR}^*$ using the surrogate metric $M$, i.e., $E_M^* = \arg\min_{e \in E'} M(\hat{\tau}_e)$. We term this *two-level model selection strategy*, and since we were more careful in selecting over hyperparameters of each meta-learner, it might help the surrogate metric in model selection.

Hence, denoting the CATE estimator population as $E = \{\cup_J E_J\}$ where $E_J$ represents all the estimators of type meta-learner $J$, the two-level selection strategy can be summarized as follows.

1. Select using meta-learner based metric ($M^J(\hat{\tau})$), $E_J^* = \arg\min_{e \in E_J} M^J(\hat{\tau}_e) \ \forall J$
2. Select using the surrogate metric $M(\hat{\tau})$, $E_M^* = \arg\min_{e \in E'} M(\hat{\tau}_e)$ where $E' = \cup_J E_J^*$

**Casual Ensembling.** The prior works typically judge the performance of any metric as per its best performing CATE estimators[2], however, this approach is prone to outliers where the top-1 choice selected using the metric is bad but the top-k choices are good. Analogous to super-learning (used successfully for predictive model selection (Ju et al., 2018)), instead of returning the best CATE estimator using a metric $M$, we instead return a weighted combination of CATE estimators, where the weight of each CATE estimator is proportional to $\exp\{\kappa M(\hat{\tau}_i)\}$, i.e. a softmax weight with $\kappa$ as the temperature which can be tuned. This helps to avoid the sharp discontinuities of the best CATE estimator selected using any surrogate metric as we select an ensemble of CATE estimators.

**Realistic benchmarks.** While the surrogate metrics themselves do not require counterfactual data, the same would be needed for the ideal PEHE score(E.q. 5) to judge the quality of the best CATE estimators returned by any metric. Hence, the prior works have experimented only with synthetic datasets where counterfactuals are known. We overcome this issue by using RealCause (Neal et al., 2020), which closely models the distribution of real datasets using state-of-the-art generative modeling techniques such as normalizing flows (Huang et al., 2018) and verifies its closeness to the original datasets using a variety of visualizations and statistical tests. They model the selection mechanism ($\mathbb{P}(W|X)$) and the output mechanism ($\mathbb{P}(Y|W, X)$) using generative models ($\mathbb{P}_{model}(W|X)$, $\mathbb{P}_{model}(Y|W, X)$), where the covariates $X$ are sampled from the observed realistic dataset. This gives us access to the interventional distributions ($\mathbb{P}_{model}(Y|do(W = 0), X)$, $\mathbb{P}_{model}(Y|do(W = 1), X)$), hence we can sample both potential outcomes in realistic datasets.

## 4 NOVEL SURROGATE CRITERIA FOR CATE MODEL SELECTION

We also propose a variety of new metrics that are based on blending ideas from other strands of the literature and which have not been examined in prior works. The primary reason for including these new metrics was to have a more comprehensive evaluation, not necessarily to beat the prior metrics.

### 4.1 ADAPTIVE PROPENSITY CLIPPING METRICS

Consider the DR Score where the pseduo-outcomes depend upon the inverse of the propensity function ($\check{\pi}(x)$). Hence, if some samples have an extremely small propensity for the observed treatment, then their pseudo-outcome estimates might be biased. Therefore, we introduce propensity clipping techniques from the policy learning and evaluation literature (Wang et al., 2017; Thomas & Brunskill, 2016; Su et al., 2019) for surrogate metrics that depend on the propensity function. We start with clipping the propensity estimate in the range $[\epsilon, 1 - \epsilon]$, $\check{\pi}(x_i) = \max\{\epsilon, \min\{1 - \epsilon, \check{\pi}(x_i)\}\}$. Then we can create a variant that uses the adaptive approach of switching to approximate $\tilde{\tau}$ as follows:

$$\tilde{\tau}_{\text{DR-Switch}} = \begin{cases} \tilde{\tau}_{\text{DR}} & \text{if } \epsilon \leq \check{\pi}_w(x) \\ \tilde{\tau}_{S/T} & \text{if } \check{\pi}_w(x) < \epsilon \end{cases} \qquad \text{(DR Switch)}$$

This metric is the same as the DR Score for samples that do not have an extremely small propensity of the observed treatment, otherwise, it uses another surrogate metric like S/T Score that doesn't depend on propensity function for reliable estimates of $\tilde{\tau}(x)$.

Another idea in policy learning is blending (Thomas & Brunskill, 2016), where we consider a convex combination of the DR pseudo-outcome and the potential-outcome based estimate, i.e. $\tilde{\tau}_i^{\text{Blend}} = \alpha\tilde{\tau}_i^{\text{IPW}} + (1 - \alpha)\tilde{\tau}_i^{S/T}$, where $\alpha$ is some constant. A successor to blending is Continuous Adaptive Blending (CAB) (Su et al., 2019), which makes $\alpha$ adaptive to the propensity of the sample and combines it with switching ideas. We present here an adaptation of CAB for CATE estimation:

$$\tilde{\tau}_{\text{DR-CAB}} = \begin{cases} \tilde{\tau}_{\text{DR}} & \text{if } \epsilon \leq \check{\pi}_w(x) \\ \frac{\check{\pi}_w(x)}{\epsilon}\tilde{\tau}_{\text{DR}} + \left(1 - \frac{\check{\pi}_w(x)}{\epsilon}\right)\tilde{\tau}_{S/T} & \text{if } \check{\pi}_w(x) < \epsilon \end{cases} \qquad \text{(DR CAB)}$$

---

[2]Recent prior work of Han & Wu (2022) also considers a variant of causal ensembling using a particular loss and based on convex regression.

## 4.2 TARGETED LEARNING

An alternative to alleviate extreme propensities is instead to learn how much inverse propensity correction we need to add. This is roughly the idea in targeted learning, which has been explored for average treatment effect (ATE) estimation, but we are not aware of any prior application for CATE estimation. We describe the *S-Learner variant of TMLE*, but the same can be done with a T-Learner. We learn a conditional linear predictor of the residual outcome $(Y - \check{\mu}(X, W))$ in a boosting manner from the inverse propensity $\check{a}(X, W) := \frac{W - \tilde{\pi}(X)}{\tilde{\pi}(X)\,(1 - \tilde{\pi}(X))}$.

$$\check{\epsilon} := \underset{f \in F}{\arg\min} \frac{1}{N} \sum_{i=1}^{N} \left(y_i - \check{\mu}(x_i, w_i) - \epsilon(x_i)\,\check{a}(x_i, w_i)\right)^2$$

The above corresponds to a weighted regression problem with weights $\check{a}(x_i, w_i)^2$ and labels $(y_i - \check{\mu}(x_i, w_i))/\check{a}(x_i, w_i)$. In our implementation, we used a causal forest approach to solve this regression problem, viewing $Y - \check{\mu}(X, W)$ as the outcome, $\check{a}(X, W)$ as the treatment, and $\epsilon(X)$ as the heterogeneous effect. Then we add the correction term to obtain the updated regression model $\check{\mu}_{\text{TMLE}}(X, W) := \check{\mu}(X, W) + \check{\epsilon}(X)\,\check{a}(X, W)$ and define the corresponding metric:

$$M^{\text{TMLE}}(\hat{\tau}) = \frac{1}{N} \sum_{i=1}^{N} \left(\hat{\tau}(x_i) - \tilde{\tau}_{\text{TMLE}}(x_i)\right)^2 \quad \tilde{\tau}_{\text{TMLE}} = \check{\mu}_{\text{TMLE}}(x, 1) - \check{\mu}_{\text{TMLE}}(x, 0) \quad \text{(TMLE Score)}$$

## 4.3 CALIBRATION SCORES

Calibration scores do not plug-in a proxy for the true $\tau(x)$, rather they check for consistency of the CATE predictions $(\hat{\tau}(x))$ within quantiles on the validation set. We split the CATE predictions $(\hat{\tau}(x))$ into $K$ percentiles (bottom 25%, next 25% etc.), and within each group $G_k(\hat{\tau})$ calculate the out-of-sample group ATE using DR pseudo-outcomes (E.q. 3) and also using the CATE predictions.

$$\text{GATE}_k^{\text{DR}}(\hat{\tau}) = \frac{1}{|G_k(\hat{\tau})|} \sum_{i \in G_k(\hat{\tau})} \tilde{\tau}_{\text{DR}}(x_i) \qquad \widehat{\text{GATE}}_k(\hat{\tau}) := \frac{1}{|G_k(\hat{\tau})|} \sum_{i \in G_k(\hat{\tau})} \hat{\tau}(x_i).$$

Viewing $\text{GATE}_k^{\text{DR}}(\hat{\tau})$ as the unbiased estimate of group ATE, we measure its weighted absolute discrepancy from the estimate of group ATE computed via input CATE predictions ($\widehat{\text{GATE}}_k(\hat{\tau})$).

$$M^{\text{Cal-DR}}(\hat{\tau}) := \sum_{k=1}^{K} |G_k(\hat{\tau})| \left| \widehat{\text{GATE}}_k(\hat{\tau}) - \text{GATE}_k^{\text{DR}}(\hat{\tau}) \right| \qquad \text{(Cal DR Score)}$$

The calibration score has been studied or RCTs in Dwivedi et al. (2020) and its variants in (Chernozhukov et al., 2018; Athey & Wager, 2019); we adapted it to be used for CATE model selection.

## 4.4 QINI SCORES

The Qini score is based on the uplift modeling literature (Surry & Radcliffe, 2011) and measures the benefit with the policy of assigning treatment based on the top-k percentile of input CATE estimates as opposed to the policy of assigning treatments uniformly at random. Let $G_{\geq k}(\hat{\tau})$ denote the group with treatment effects in the top $k$-th percentile of the input CATE estimates. We can measure the group ATE for it using DR pseudo-outcomes (E.q. 3), $\text{GATE}_{\geq k}^{\text{DR}}(\hat{\tau}) := \frac{1}{|G_{\geq k}(\hat{\tau})|} \sum_{i \in G_{\geq k}} \tilde{\tau}_{\text{DR}}(x_i)$.

The cumulative effect from this group should be much better than treating the same population uniformly at random, which can be approximated as $\text{ATE}^{\text{DR}} := \frac{1}{N} \sum_{i=1}^{N} \tilde{\tau}_{\text{DR}}(x_i)$. This yields the following score (higher is better):

$$M^{\text{Qini-DR}}(\hat{\tau}) := \sum_{k=1}^{100} |G_{\geq k}(\hat{\tau})| \left(\text{GATE}_{\geq k}^{\text{DR}}(\hat{\tau}) - \text{ATE}^{\text{DR}}\right) \qquad \text{(Qini DR Score)}$$

## 5 EMPIRICAL ANALYSIS

We now present our findings from the extensive benchmarking study of *34 metrics* for selecting amongst a total of *415 CATE estimators* across *78 datasets* over *20 random seeds* for each dataset.

### 5.1 EXPERIMENT SETUP

We work with the ACIC 2016 (Dorie et al., 2019) benchmark, where we discard datasets that have variance in true CATE lower than $0.01$ to ensure heterogeneity; which leaves us with 75 datasets from the ACIC 2016 competition. Further, we incorporate three realistic datasets, LaLonde PSID, LaLonde CPS (LaLonde, 1986), and Twins (Louizos et al., 2017), using RealCause. For each dataset, the CATE estimator population comprises 7 different types of meta-learners, where the nuisance models ($\hat{\eta}$) are learned using AutoML (Wang et al., 2021). For the CATE predictor ($\hat{f}$) in direct meta-learners, we allow for multiple choices with variation across the regression model class and hyperparameters, resulting in a diverse collection of estimators for each direct meta-learner. Even the most recent benchmarking study by Curth & van der Schaar (2023) did not consider a large range of hyperparameters for direct meta-learners, while me make the task of model selection more challenging with a larger grid of hyperparameters. For the set of surrogate metrics, we incorporate all the metrics used in the prior works and go beyond to consider various modifications of them, along with the novel metrics described in Section 4. As stated before in Section 3, we use AutoML for selecting the nuisance models ($\check{\eta}$) of surrogate metrics on the validation set. More details regarding the experiment setup can be found in Appendix C.

### 5.2 RESULTS

Following the discussion in Section 3, we compute PEHE of the best estimators selected by a surrogate metric to judge its performance, PEHE($E_M^*$)= $\frac{1}{|E_M^*|} * \sum_{e \in E_M^*} L(\hat{\tau}_e)$. Since the scale of the true CATE can vary a lot across datasets, we compute a normalized version where we take the % difference of the PEHE of the best estimators chosen by each metric ($E_M^*$) from the PEHE of the overall best estimator ($E^\dagger$), Normalized-PEHE($M$)= [ PEHE($E_M^*$) - PEHE($E^\dagger$) ] / PEHE($E^\dagger$). For each dataset, we report the mean (standard error) Normalized-PEHE over 20 random seeds. Since we have multiple datasets under the ACIC 2016 benchmark, we first compute the mean performance across them and then compute the mean and standard error across the random seeds. For each dataset group, we **bold** the dominating metrics using the following rule; A metric $M$ is said to be a dominating metric if the confidence interval of the performance of metric $M$ either overlaps or lies strictly below the confidence interval of the performance of any other metric $\tilde{M} \neq M$.

#### 5.2.1 SINGLE-LEVEL MODEL SELECTION STRATEGY

We first provide results with the single-level model selection strategy for a selected list of metrics in Table 1. Results with the complete list of surrogate metrics can be found in Table 7 in Appendix D.

**Doubly Robust and TMLE variants as globally dominating metrics.** Across all the datasets, DR T Score (and its variants) and TMLE T score are optimal as compared to the other metrics. They produce even better results than Calibration and Qini based scores. Further, the improvements due to adaptive propensity clipping techniques (Switch, CAB) over the basic DR score are not significant.

**Plug-in surrogate metrics are globally optimal.** It is interesting to observe that plug-in metrics like T/X Score are rarely dominated by other metrics! This highlights the importance of learning nuisance models with AutoML, as it enhances the model selection ability due to lower bias in the estimated nuisance parameters. Since the prior works did not search over a large grid for learning nuisance models, that could explain why the plug-in metrics were sub-optimal in their results.

**Superior performance of T-Learner based metrics.** In Table 2 we compare the metrics that have the choice of estimating the potential outcomes ($\hat{\mu}_0, \hat{\mu}_1$) using either S-Learner or T-Learner. We find that metrics with the T-Learner strategy are better than those with S-Learner strategy in all the cases, which further highlights that choice of nuisance models is critical to the performance of surrogate metrics.

| Metric | ACIC 2016 | LaLonde CPS | LaLonde PSID | TWINS |
|---|---|---|---|---|
| Value Score | 1.05$e$+7 (4.31$e$+6) | 6.63 (5.52) | **0.48** (**0.06**) | 0.57 (0.15) |
| Value DR Score | 13.02 (11.73) | 2.33 (1.41) | **0.46** (**0.05**) | 1.61 (1.02) |
| Match Score | 3.60 (0.16) | **0.23** (**0.04**) | 0.50 (0.06) | 0.38 (0.08) |
| S Score | 0.95 (0.02) | 0.90 (0.04) | 0.74 (0.04) | **0.29** (**0.05**) |
| T Score | **0.56** (**0.02**) | **0.16** (**0.03**) | **0.42** (**0.03**) | **0.31** (**0.05**) |
| X Score | **0.56** (**0.02**) | **0.16** (**0.03**) | **0.41** (**0.03**) | 0.35 (0.06) |
| R Score | 4.0 (0.11) | 0.83 (0.04) | 0.67 (0.03) | 0.60 (0.11) |
| Influence Score | 1455.75 (1439.46) | 0.95 (0.04) | 0.80 (0.02) | 1.08 (0.1) |
| IPW Score | 3.21 (0.12) | **0.25** (**0.05**) | **0.32** (**0.02**) | 0.37 (0.06) |
| DR T Score | **0.56** (**0.02**) | **0.16** (**0.02**) | **0.41** (**0.03**) | **0.32** (**0.07**) |
| DR Switch T Score | **0.56** (**0.02**) | **0.16** (**0.03**) | **0.41** (**0.03**) | **0.28** (**0.05**) |
| DR CAB T Score | **0.56** (**0.02**) | **0.16** (**0.03**) | **0.41** (**0.03**) | **0.33** ( **0.06** ) |
| TMLE T Score | **0.64** (**0.03**) | **0.16** (**0.03**) | **0.42** (**0.03**) | **0.31** (**0.05**) |
| Cal DR T Score | 3.45 (0.11) | **0.17** (**0.03**) | **0.42** (**0.03**) | **0.21** (**0.03**) |
| Qini DR T Score | 1.32 (0.07) | 2.87 (1.53) | 0.57 (0.05) | 2.08$e$+7 (1.90$e$+7) |

Table 1: Normalized PEHE of the **best estimators** chosen by each metric with the **single-level model selection strategy**; results report the mean (standard error) across 20 seeds and also across datasets for the ACIC 2016 benchmark. **Lower value is better.**

| Metric | ACIC 2016 | LaLonde CPS | LaLonde PSID | TWINS |
|---|---|---|---|---|
| S Score | 0.95 (0.02) | 0.90 (0.04) | 0.74 (0.04) | **0.29** (**0.05**) |
| T Score | **0.56** (**0.02**) | **0.16** (**0.03**) | **0.42** (**0.03**) | **0.31** (**0.05**) |
| DR S Score | 0.93 (0.02) | 0.85 (0.05) | 0.73 (0.04) | 0.35 (0.06) |
| DR T Score | **0.56** (**0.02**) | **0.16** (**0.02**) | **0.41** (**0.03**) | **0.32** (**0.07**) |
| TMLE S Score | 1.06 (0.04) | 0.91 (0.04) | 0.74 (0.04) | **0.26** (**0.05**) |
| TMLE T Score | **0.64** (**0.03**) | **0.16** (**0.03**) | **0.42** (**0.03**) | **0.31** (**0.05**) |
| Cal DR S Score | 5.78 (0.19) | 0.87 (0.05) | 0.72 (0.04) | **0.19** (**0.03**) |
| Cal DR T Score | 3.45 (0.11) | **0.17** (**0.03**) | **0.42** (**0.03**) | **0.21** (**0.03**) |

Table 2: Comparing the S-Learner vs T-Learner based metrics. Each cell represents the Normalized PEHE of the **best estimators** with the **single-level strategy**; results report the mean (standard error) across 20 seeds and also across datasets for the ACIC 2016 benchmark. **Lower value is better.**

### 5.2.2 TWO-LEVEL MODEL SELECTION STRATEGY

We now provide results with the two-level model selection strategy for a selected list of metrics in Table 3. Results with the complete list of metrics can be found in Table 8 in Appendix D.

**Better performance than single-level strategy.** We find that the two-level selection strategy performs much better as compared to the single-level selection strategy, and we find better performance in approximately **28.7**% cases over all datasets and metrics; with statistically indistinguishable performance for the other cases. Since in no scenario it happens that this strategy gets dominated by the single-level selection strategy, we recommend this as a good practice for CATE model selection.

In fact, Qini DR score ended up as a dominating metric for almost all of the datasets with the two-level strategy, while it was among the worst metrics with the single-level strategy for the TWINS dataset. Also, the Value DR score ends up as a globally dominating metric with this strategy, which is a big improvement in contrast to its performance before. Further, the major conclusions from

| Metric | ACIC 2016 | LaLonde CPS | LaLonde PSID | TWINS |
|--------|-----------|-------------|--------------|-------|
| Value Score | 3.97 (1.98) | **0.34 (0.09)** | **0.43 (0.03)** | **0.21 (0.03)** |
| Value DR Score | **0.64 (0.03)** | **0.25 (0.08)** | **0.47 (0.04)** | **0.21 (0.03)** |
| Match Score | 1.76 (0.09) | **0.17 (0.03)** | **0.45 (0.03)** | **0.21 (0.03)** |
| S Score | 0.93 (0.02) | 0.90 (0.04) | 0.75 (0.04) | **0.21 (0.03)** |
| T Score | **0.56 (0.02)** | **0.16 (0.03)** | **0.41 (0.03)** | **0.21 (0.03)** |
| X Score | **0.56 (0.02)** | **0.16 (0.03)** | **0.41 (0.03)** | **0.21 (0.03)** |
| R Score | 3.88 (0.11) | 0.86 (0.03) | 0.62 (0.03) | **0.21 (0.03)** |
| Influence Score | 3.26 (0.1) | 0.93 (0.04) | 0.77 (0.03) | **0.16 (0.02)** |
| IPW Score | 1.41 (0.06) | **0.16 (0.04)** | **0.38 (0.02)** | **0.21 (0.03)** |
| DR T Score | **0.56 (0.02)** | **0.16 (0.02)** | **0.41 (0.03)** | **0.21 (0.03)** |
| DR Switch T Score | **0.56 (0.02)** | **0.16 (0.03)** | **0.41 (0.03)** | **0.21 (0.03)** |
| DR CAB T Score | **0.56 (0.02)** | **0.16 (0.03)** | **0.41 (0.03)** | **0.21 (0.03)** |
| TMLE T Score | **0.61 (0.03)** | **0.16 (0.03)** | **0.42 (0.03)** | **0.21 (0.03)** |
| Cal DR T Score | **0.62 (0.02)** | **0.19 (0.04)** | **0.42 (0.03)** | **0.22 (0.03)** |
| Qini DR T Score | **0.58 (0.02)** | **0.14 (0.03)** | 0.52 (0.03) | **0.24 (0.04)** |

Table 3: Normalized PEHE of the **best estimators** chosen by each metric with the **two-level model selection strategy**; results report the mean (standard error) across 20 seeds and also across datasets for the ACIC 2016 benchmark. **Lower value is better.**

before regarding the dominance of DR/TMLE and the plug-in T/X metrics are still valid with the two-level strategy, along with the superior performance of T-Learner over S-Learner based metrics.

Hence, a two-level selection strategy can lead to significant benefits, and designing better methods towards the same can be a fruitful direction. Note the proposed choice of using meta-learner based metric ($M^J(\hat{\tau})$) to select amongst all meta-estimators of type $J$ is not guaranteed to be optimal, and we chose it to mimic the inductive bias of meta-learner $J$. In Appendix D, Tables 10 to 13 provide results for selecting amongst only a particular class of meta-learners using any surrogate metric, and we can see that in some cases the optimal choice is not $M^J(\hat{\tau})$. E.g., in Table 13, the S Score is not always optimal for selecting amongst estimators of type Projected S-Learner.

**Enhanced performance with causal ensembling.** Since we are still selecting the best meta-learner with the two-level strategy in Table 3, we now consider selecting an ensemble of meta-learners with the two-level strategy and provide its results in Table 9 (Appendix D). We find that ensembling is statistically better than non-ensembling on $\approx$ **5.8%** of the experiments (across all datasets and metrics), and otherwise has statistically indistinguishable performance.

## 6 CONCLUSION

Our work shows the importance of consistent evaluation across a wide range of datasets for surrogate model selection metrics, which leads to more detailed trends as opposed to prior works. With well-tuned nuisance models via AutoML, we show that even plug-in surrogate metrics (T Score) can be competitive for model selection. Further, we present novel strategies of two-level model selection and causal ensembling, which can be adopted to enhance the performance of any surrogate metric. Among all the metrics, the DR/TMLE based variants always seem to be among the dominating metrics, hence if one were to use a global rule, such metrics are to be preferred. However, we believe that a more contextual metric is the right avenue and has great potential for future research.

ACKNOWLEDGEMENTS

The authors would like to thank the reviewers for their detailed feedback and suggestions! We also thank Amit Sharma for helpful pointers regarding the presentation of the work. The experiments were enabled in part by computational resources provided by Calcul Québec (`calculquebec.ca`) and the Digital Research Alliance of Canada (`alliancecan.ca`). Ioannis Mitliagkas acknowledges support by an NSERC Discovery grant (RGPIN-2019-06512), a Microsoft Research collaborative grant and a Canada CIFAR AI chair.

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

# Appendix

## LIST OF CONTENTS

The content in the Appendix has been organized as follows.

# A    REVIEW OF CATE ESTIMATION

## A.1    IDENTIFICATION OF CATE

The objective of CATE by definition relies on interventional data, which implies we cannot estimate this using only observational data.

$$\text{CATE:} \quad \tau(x) = \mathbb{E}[Y(1) - Y(0)|X = x] = \mu_1(x) - \mu_0(x).$$

However, we can make the ignorability assumptions (Peters et al., 2017) which enables us to identify the expected potential outcomes from observational data ($E[Y(t)|X = x] = E[Y|W = t, X = x]$), hence identify CATE from observational data.

The ignorability assumption consists of the following three assumptions:

- **Consistency:** It implies that the outcome we observe reflects the true potential outcome under the observed treatment, $Y = W \cdot Y(1) + (1 - W) \cdot Y(0)$.

- **Exchangeability:** It implies that we do not have any unobserved confounders, $Y(0), Y(1) \perp\!\!\!\perp W \mid X$. Unobserved confounders would lead to open backdoor paths between Y(0), Y(1) and W, hence we would not have conditional independence.

- **Overlap:** It implies that the treatment assignment for each sample is probabilistic, $0 < \pi(x) < 1 \;\; \forall x \in X$. Therefore, each sample has a non-trivial chance of being assigned to either group ($W = 0$ or $W = 1$).

## A.2    META-LEARNERS FOR CATE ESTIMATION

We consider only meta-learners for CATE estimation as they reduce CATE estimation to a series of (weighted) regression and classification problems, which makes it easy to apply out-of-the-box machine learning techniques to solve each sub-problem. Such approaches have been extensively studied in the literature and are heavily used in industry practice. Following (Curth & Van der Schaar, 2021), we divide the meta-learners into two categories, indirect and direct meta-learners.

**Indirect Meta-Learners.**    The main learning objective of these estimators is to accurately model the potential outcomes $(\hat{\mu}_0, \hat{\mu}_1)$ using the observational data and then we can obtain CATE estimate indirectly as the difference in the learned potential outcomes.

The **T-Learner** approach approximates the potential outcome $\mathbb{E}[Y|W = 0, X = x]$ as $\hat{\mu}_0$ by regressing $Y$ on $X$ using samples from the un-treated population $\{x, y\}_{w=0}$, and $\mathbb{E}[Y|W = 1, X = x]$ as $\hat{\mu}_1$ by regressing $Y$ on $X$ using samples $\{x, y\}_{w=1}$ from the treated population.

$$\hat{\tau}_T(x) = \hat{\mu}_1(x) - \hat{\mu}_0(x) \tag{7}$$

While the **S-Leaner** approach learns a single regression model $\hat{\mu}(x, w)$, regressing $Y$ jointly on the features $X$ and the treatment assignments $W$ from observational data $\{x, w, y\}$.

$$\hat{\tau}_S(x) = \hat{\mu}(x, 1) - \hat{\mu}(x, 0) \tag{8}$$

If the nuisances models $(\hat{\mu}_0(x), \hat{\mu}_1(x), \hat{\mu}(x, w))$ are universal function approximators, then S-Learner and T-Leaner would offer unbiased estimates of CATE. However, there will be statistical errors in practice due to the nuisance model class not capturing the true potential outcome. This will be amplified more as compared to standard regression tasks as we also do model-extrapolation while computing CATE, i.e. use $\hat{\mu}_w(x)$ to make predictions for data points with the same features as observed data points but different treatment assignment (Kennedy, 2020).

**Direct Meta-Learners.**    Direct meta-learners also learn the nuisance parameters $(\hat{\mu}, \hat{\pi})$ from observational data but then learn a regression function to directly predict CATE. For example, we can transform the indirect S-Learner described above into a direct meta-learner, known as the **Projected S-Learner** (Battocchi et al., 2019), which learns a final projection step by regressing the difference in the estimated potential outcomes $\hat{\mu}(x, 1) - \hat{\mu}(x, 0)$ on the covariates.

$$\hat{\tau}_{PS} = \arg\min_{f \in F} \sum_{\{x,w,y\}} \left( \hat{\mu}(x,1) - \hat{\mu}(x,0) - f(x) \right)^2 \tag{9}$$

This introduces an extra regularization step which could avoid potential overfitting, especially if we have some prior belief on the functional form or the smoothness of the CATE function.

Another example is the the **X-Learner** (Künzel et al., 2019) approach, which first trains the nuisance models $(\hat{\mu}_0(x), \hat{\mu}_1(x))$ as in the T-Learner approach. But rather than using the difference in estimated potential outcomes to predict CATE, we model the CATE directly for the control and the treatment group using regression models $(f_\theta^0, f_\theta^1)$ as follows.

$$
\begin{aligned}
\hat{\tau}^0 &= \arg\min_{f \in F} \sum_{i, w_i = 0} \left( \hat{\mu}_1(x_i) - y_i - f(x_i) \right)^2 \\
\hat{\tau}^1 &= \arg\min_{f \in F} \sum_{i, w_i = 1} \left( y_i - \hat{\mu}_0(x_i) - f(x_i) \right)^2
\end{aligned} \tag{10}
$$

The first approach imputes treatment counterfactuals for each control sample and views $(\hat{\mu}_1(x) - y)$ as a proxy for the individual treatment effect. Then learn the CATE predictor by regressing that on covariates $x$. The second approach takes the analogous approach using the treated population. Finally, we use the learned propensity model $\hat{\pi}(x)$ to combine the CATE predictions from both groups:

$$\hat{\tau}_X(x) = \hat{\pi}(x)\,\hat{\tau}^0(x) + (1 - \hat{\pi}(x))\,\hat{\tau}^1(x) \tag{11}$$

The intuition behind the weighting is that the estimate $\hat{\tau}^0$ relies on the function $\hat{\mu}_1$, which performs well in regions of $X$ where we have many treated units (i.e. when $\pi(x)$ is large). Similarly, the estimate $\hat{\tau}^1$ relies on $\hat{\mu}_0$ which performs well in regions where we have many controls units.

We now discuss in detail the widely used direct meta-learners; doubly robust and double machine learning techniques for CATE estimation.

**Doubly Robust Learner (DR-Learner)**   To avoid the heavy dependence on the potential outcome regression functions with indirect meta-learners, recent works have proposed generalizations of the doubly robust estimator (traditionally used for ATE inference), for CATE estimation (Foster & Syrgkanis, 2019; Kennedy, 2020). The DR learner is a mixture of the S-learner with an inverse propensity (IPW) based approach. First, we state the vanilla inverse propensity based learner and then the DR leaner The inverse propensity weighted (IPW) estimator, uses the learned propensity model $(\hat{\pi}_w(x))$ to compute the IPW pseudo-outcome as follows:

$$y^{\text{IPW}}(\hat{\eta}) = y_1^{\text{IPW}}(\hat{\eta}) - y_0^{\text{IPW}}(\hat{\eta}) \quad \text{where} \quad y_w^{\text{IPW}}(\hat{\eta}) = \frac{y}{\hat{\pi}_w(x)} \tag{12}$$

It can be shown that under the true propensity model, we have the following (Peters et al., 2017):

$$\mathbb{E}[Y_w^{\text{IPW}}(\eta)|X] = \mathbb{E}[Y(w)|X]$$

Hence, $Y_w^{\text{IPW}}(\hat{\eta})$ gives an unbiased estimate of the potential outcome $Y(t)$ and the pseudo-outcome $Y^{\text{IPW}}(\hat{\eta}) = Y_1^{\text{IPW}}(\hat{\eta}) - Y_0^{\text{IPW}}(\hat{\eta})$ can be used to approximate the CATE. Therefore, we can learn the CATE predictor $(\hat{f}_{\text{IPW}})$ by regressing the IPW pseudo outcomes on the covariates.

$$\hat{\tau}_{\text{IPW}} := \arg\min_{f \in F} \sum_{\{x,y,w\}} \left( y_w^{\text{IPW}}(\hat{\eta}) - f(x) \right)^2 \tag{13}$$

Unlike the S-Learner, the IPW-Learner does not depend on any potential outcome regression model but heavily relies on the learned propensity model $\hat{\pi}_w$. Further, there are numerical stability issues if the true probability of a certain treatment is low, as then we are dividing by very small numbers. Hence, the IPW-Learner can have very high variance, which can be overcome by doubly robust learning (Foster & Syrgkanis, 2019; Kennedy, 2020). It can be interpreted as a hybrid approach,

where it first directly approximates the potential outcomes ($\hat{\mu}(x, w)$) like S-Learner, and then it debiases the estimate using the IPW based approach on the residual ($y - \hat{\mu}(x, w)$), as shown below:

$$y^{\text{DR}}(\hat{\eta}) = y_1^{\text{DR}}(\hat{\eta}) - y_0^{\text{DR}}(\hat{\eta}) \quad \text{where} \quad y_w^{\text{DR}}(\hat{\eta}) = \hat{\mu}(x, w) + \frac{y - \hat{\mu}(x, w)}{\hat{\pi}_w(x)} \tag{14}$$

Finally, we learn the CATE predictor ($\hat{f}_{\text{DR}}$) that maps the covariates to the DR pseudo-outcomes:

$$\hat{\tau}_{\text{DR}} := \arg\min_{f \in F} \sum_{\{x,w,y\}} \left( y^{\text{DR}}(\hat{\eta}) - f(x) \right)^2 \tag{15}$$

**Double Machine Learning (R-Learner)**    The Dobule ML approach to CATE estimation (Chernozhukov et al., 2016; Nie & Wager, 2021), also known in the literature as the R-Learner (Nie & Wager, 2021), solves the CATE estimation task by learning the following nuisance models ($\hat{\pi}, \hat{q}$), where $\hat{\pi}$ is an estimate of the true propensity model, and $\hat{q}$ is an estimate of $\mathbb{E}[Y|X]$, i.e, predicting potential outcomes only from the covariates. Then it computes the residuals from both the potential outcome prediction task ($\tilde{y} := y - \hat{\mu}(x)$) and the treatment assignment task ($\tilde{w} := w - \hat{\pi}(x)$). In the final step it learns the CATE predictor ($\hat{f}_{\text{DML}}$) by minimizing the following loss function:

$$\hat{\tau}_{\text{DML}} := \arg\min_{f \in F} \sum_{\{x,y,w\}} (\tilde{y} - f(x) \cdot \tilde{w})^2 = \arg\min_{f \in F} \sum_{\{x,y,w\}} \tilde{w}^2 \left( \tilde{y}/\tilde{w} - f(x) \right)^2 \tag{16}$$

This can be viewed as a weighted regression problem with weights $\tilde{w}_i^2$ and labels $\tilde{y}_i/\tilde{w}_i$. This learning objective can be justified by simply observing that for a binary treatment, we have:

$$\mathbb{E}[Y \mid X, W] = \tau(X) \cdot W \Rightarrow \mathbb{E}[Y - \mathbb{E}[Y \mid X] \mid X, W] = T(X) \cdot (W - \mathbb{E}[W \mid X])$$

Therefore, the R-Learner loss is the square loss that corresponds to the latter regression equation. The R-Learner has been shown to enjoy theoretical robustness properties similar to the DR learner (i.e. that the impact of errors in the nuisance models $\hat{\pi}, \hat{\mu}$) on the final estimate is alleviated, albeit for the R-Learner the accuracy of the propensity is more important and the method is not consistent if the propensity model is not consistent (while the DR-Learner would be if the regression model was consistent but the propensity model was inconsistent). Moreover, if the model space $F$ does not contain the true CATE model $\tau$, then the outcome of the DR Learner can be interpreted asymptotically as a projection of $\tau$ on the model space, with respect to the $L_2$ norm, while the outcome of the R-Learner is a weighted projection, weighted by the variance of the treatment; hence approximating the true $\tau$ more accurately in regions where the treatment is more random.

## B  REVIEW OF CATE MODEL SELECTION SURROGATE METRICS

**Surrogate PEHE metrics.**   Majority of the surrogate metrics for CATE model selection that we consider in our analysis learn an approximation of the ground truth CATE as $\tilde{\tau}(x)$ and then compute the PEHE (E.q. 5), i.e, mean squared loss of the input CATE estimator's prediction ($\hat{\tau}(x)$) from $\tilde{\tau}(x)$ on the validation set.

$$M(\hat{\tau}) = \frac{1}{N} \sum_{i=1}^{N} (\hat{\tau}(x_i) - \tilde{\tau}(x_i))^2 \tag{17}$$

Since the approximate true CATE ($\tilde{\tau}(x)$) is, in general, a function of nuisance parameters as well, to differentiate the nuisance parameters of the surrogate metrics ($M$) as compared to those associated with the CATE estimators ($E$), we represent the nuisance models of the metrics with downward hats ($\check{\mu}, \check{\pi}$), while nuisance models of the estimators have upward hats ($\hat{\mu}, \hat{\pi}$). Note that the nuisance models associated with the surrogate metrics are trained only on the validation set and we do not assume any access to training data in this stage. These metrics are broadly classified into two categories, *plug-in surrogate* and *pseudo-outcome surrogate* metrics, as discussed below.

**Plug-in surrogate metrics.** The plug-in surrogate metrics learn $\tilde{\tau}(x)$ in an analogous way to meta-learners and use it to score the input CATE estimate ($\hat{\tau}(x)$). For example, we can construct the plug-in metrics based on the indirect meta-learner strategies as follows, which were also used as baselines by  Alaa & Van Der Schaar (2019).

$$M^S(\hat{\tau}) := \frac{1}{N} \sum_{i=1}^{N} (\hat{\tau}(x_i) - \tilde{\tau}_S(x_i))^2 \qquad \tilde{\tau}_S(x) := \check{\mu}(x, 1) - \check{\mu}(x, 0) \qquad (S \text{ Score})$$

$$M^T(\hat{\tau}) := \frac{1}{N} \sum_{i=1}^{N} (\hat{\tau}(x_i) - \tilde{\tau}_T(x_i))^2 \qquad \tilde{\tau}_T(x) := \check{\mu}_1(x) - \check{\mu}_0(x) \qquad (T \text{ Score})$$

Another plug-in surrogate metric we consider is the *matching score*, which estimates $\tilde{\tau}(x)$ using the matching method (Rolling & Yang, 2013), for each point $x_i$, find its nearest neighbour from the opposite treatment group, $\tilde{i} = \arg\min_{j|w_j \neq w_i} \|x_j - x_i\|^2$, then define $\tilde{\tau}_i = y_i - y_{\tilde{i}}$.

$$M^{\text{Match}}(\hat{\tau}) := \frac{1}{N} \sum_{i=1}^{N} (\hat{\tau}(x_i) - \tilde{\tau}_{\text{Match}}(x_i))^2 \qquad \tilde{\tau}_{\text{Match}}(x_i) := y - y_{\tilde{i}} \qquad (\text{Matching Score})$$

Note that in general, we could use direct meta-learners as well for estimating $\tilde{\tau}(x)$ in plug-in metrics, however, we do not consider them as they are more susceptible to congeniality bias (Curth & van der Schaar, 2023). For surrogate metrics based on direct meta-learner strategies, we only consider the pseudo-outcome variant as described ahead.

**Pseudo-outcome surrogate metrics.** These surrogate metrics approximate $\tilde{\tau}(x)$ using pseudo-outcomes $Y(\check{\eta})$ such that under the assumption of true nuisance parameters, we have $\mathbb{E}[Y_\eta|X] = \tau(x)$, hence justifying their choice for approximating the ground truth CATE.

For example, we can construct the pseudo-outcome metrics using IPW (E.q. 12) and DR pseudo-outcome (E.q. 14), which was first proposed for model selection by  Saito & Yasui (2020).

$$M^{\text{IPW}}(\hat{\tau}) := \frac{1}{N} \sum_{i=1}^{N} (\hat{\tau}(x_i) - \tilde{\tau}_{\text{IPW}}(x_i))^2 \qquad \tilde{\tau}_{\text{IPW}} := y^{\text{IPW}}(\check{\eta}) \qquad (\text{IPW Score})$$

$$M^{\text{DR}}(\hat{\tau}) := \frac{1}{N} \sum_{i=1}^{N} (\hat{\tau}(x_i) - \tilde{\tau}_{\text{DR}}(x_i))^2 \qquad \tilde{\tau}_{\text{DR}} := y^{\text{DR}}(\check{\eta}) \qquad (\text{DR Score})$$

Since we use the S-Learner strategy to obtain the nuisance models for computing the DR pseudo outcomes, we will refer to it as **DR S Score**. If instead we used the T-Learner strategy ($\check{\mu}_0, \check{\mu}_1$) for computing DR pseudo outcomes, $y_w^{\text{DR}}(\check{\eta}) = \check{\mu}_w(x) + \frac{y - \check{\mu}_w(x)}{\check{\pi}_w(x)}$, we will refer to it as **DR T Score**.

Further, we define **propensity clipped versions** of metrics that depend on the propensity function. We update the propensity function with the clipped propensity estimates in the range $[\epsilon, 1 - \epsilon]$:

$$\tilde{\pi}(x_i) = \max \left\{ \epsilon, \min \left\{ 1 - \epsilon, \check{\pi}(x_i) \right\} \right\} \tag{18}$$

Whenever such propensity clipping is introduced in the score, we will annotate it with the extra keyword "Clip" (e.g. *IPW Clip Score*, *DR Clip T Score*, etc.).

We also consider **R-Score** (Nie & Wager, 2021) that uses Double Machine Learning (DML) to approximate $\tilde{\tau}(x)$, which was found to be the best performing surrogate metric in the evaluation study of Schuler et al. (2018). Using the approximation of $\tilde{\tau}(X)$ via the DML method, we know that $\tilde{\tau}(X)$ would be the solution to the regression problem (E.q. 16). Hence, to compute the deviation of the input CATE estimator's prediction $\hat{\tau}(x)$ from $\tilde{\tau}(x)$, we substitute $\hat{\tau}(x)$ in the DML regression problem (equation 16), as shown below, where $\check{\pi}(x_i)$ is the probability of $x_i$ belonging to the treatment class $w = 1$, i.e., the propensity.

$$M^R(\hat{\tau}) := \frac{1}{N} \sum_{i=1}^{N} \left( (y_i - \check{\mu}(x_i)) - \hat{\tau}(x_i) \left( w_i - \check{\pi}(x_i) \right) \right)^2 \tag{R-Score}$$

The **Influence Score** proposed by Alaa & Van Der Schaar (2019) also falls in the category of pseudo-outcome surrogate metrics. To compute the influence score, we first compute a plug-in surrogate metric using T-Learner (E.q. $T$ Score) and then debias it using influence functions. Following Theorem 2 in their work, the correction term is defined as follows:

$$\text{IF-Correction}(\hat{\tau}, \tilde{\tau}) = (1 - B)(\tilde{\tau}(X))^2 + BY(\tilde{\tau}(X) - \hat{\tau}(X)) - (A + 1)\tilde{L}(\hat{\tau}) + (\hat{\tau}(X))^2$$

where $A = W - \check{\pi}(X)$, $B = 2\,W\,A\,C^{-1}$, and $C = \check{\pi}(X)\,(1 - \check{\pi}(X))$ This is summarized in the equation below, where we add influence correction to the plug-in surrogate metric.

$$M^{\text{IF}}(\hat{\tau}) := \frac{1}{N} \sum_{i=1}^{N} \left[ (\hat{\tau}(x_i) - (\check{\mu}_1(x_i) - \check{\mu}_0(x_i)))^2 \right] + \text{IF-Correction}(\hat{\tau}, \tilde{\tau}) \tag{Influence Score}$$

Finally, we also propose the **X-Score** that computes pseudo-outcomes based on the X-Leaner methodology (E.q. 11). Following the learning objective of CATE predictors per group (E.q. 10), the potential outcomes associated with each group are given as follows:

$$\tilde{\tau}^0(x) = \check{\mu}_1(x) - y \qquad \tilde{\tau}^1(x) = y - \check{\mu}_0(x)$$

These are combined using the propensity model $(\check{\pi}(x))$ to construct the final potential outcome based approximation of the true CATE as $\tilde{\tau}(x) = \check{\pi}(x)\tilde{\tau}^0(x) + (1 - \check{\pi}(x))\tilde{\tau}^1(x)$

$$M^X(\hat{\tau}) := \frac{1}{N} \sum_{i=1}^{N} (\hat{\tau}(x_i) - \tilde{\tau}_X(x_i))^2 \quad \tilde{\tau}_X(x) := \check{\pi}(x)\tilde{\tau}^0(x) + (1 - \check{\pi}(x))\tilde{\tau}^1(x) \tag{X Score}$$

**Utility of treatment policy based metrics.** As opposed to approximating the true CATE, we also consider the surrogate metrics based on the estimates of the value of an optimal policy designed via the input estimator's CATE prediction $(\hat{\tau}(x))$. We derive the optimal treatment policy corresponding to the input CATE estimator as $\hat{d}(x) := \mathbb{1}(\hat{\tau}(x) > 0)$ (assuming larger potential outcome is better). Then, one such metric, *value score* (Zhao et al., 2017), constructs an IPW (Equation 12) based estimates of the policy value:

$$M^{\text{value}}(\hat{\tau}) := \frac{1}{N} \sum_{i=1}^{N} \frac{y_i}{\tilde{\pi}_{w_i}(x_i)} \cdot \mathbb{1}(w_i = \hat{d}(x_i))) \tag{Value Score}$$

An issue with the Value Score is that it only utilizes a portion of the data where the decision policy and the observed treatment agree. This is handled in the *value DR score* (Athey & Wager, 2017), where we use the DR pseudo-outcomes (Equation 14), to estimate the policy value:

$$M^{\text{value-DR}}(\hat{\tau}) := \frac{1}{N} \sum_{i=1}^{N} \hat{d}(x_i) \cdot y_{w_i}^{\text{DR}}(\check{\eta}) \tag{Value DR Score}$$

**Proposed surrogate metrics.** In Section 4 we introduced novel surrogate metrics based on ideas from related areas like policy learning, uplift modeling, etc. The primary reason for including these new metrics was to have a more comprehensive evaluation and include more good candidates for model selection, not necessarily to beat the prior metrics. The ideas of switching, blending, etc. have been well-studied and extensively used in the policy learning literature, but their application in the context of CATE estimation/ model selection has not been shown. Similarly, targeted learning has been used for learning ATE estimators but it remains unexplored for CATE model selection.

The motivation for adaptive propensity clipping and targeted learning metrics is the extreme propensity regime, where the metrics that depend on inverse propensity scores (IPW, DR) can become biased. Hence, these proposed metrics can handle the extreme propensity region (very small propensity for the observed treatment) by using estimates from the regression-based learners (S/T Learner) as in DR Switch, DR CAB, or adding the inverse propensity correction as in TMLE Score. Since we only described the adaptive propensity clipped variants of DR-Score in the main text, we now provide the same for the IPW-Score for convenience.

$$\tilde{\tau}_{\text{IPW-Switch}} = \begin{cases} \tilde{\tau}_{\text{IPW}} & \text{if } \epsilon \leq \check{\pi}_w(x) \\ \tilde{\tau}_{S/T} & \text{if } \check{\pi}_w(x) < \epsilon \end{cases} \qquad \text{(IPW Switch)}$$

$$\tilde{\tau}_{\text{IPW-CAB}} = \begin{cases} \tilde{\tau}_{\text{IPW}} & \text{if } \epsilon \leq \check{\pi}_w(x) \\ \frac{\check{\pi}_w(x)}{\epsilon} \tilde{\tau}_{\text{IPW}} + \left(1 - \frac{\check{\pi}_w(x)}{\epsilon}\right) \tilde{\tau}_{S/T} & \text{if } \check{\pi}_w(x) < \epsilon \end{cases} \qquad \text{(IPW CAB)}$$

For the Calibration score, the intuition is to match the group ATE across subgroups denoted by the different percentiles of the CATE estimates. We compute the group ATE using DR/TMLE which is viewed as an unbiased sample of the group ATE. Subsequently, we compute the weighted absolute error of the group ATE computed using the input CATE estimates against the unbiased group ATE. For the Qini score, rather than computing the group ATE with the input CATE estimates as in Calibration, here we compare the improvement with unbiased group ATE estimates (computed using DR/TMLE) versus the uniform sampling estimate. Intuitively, if there is heterogeneity in the data, then we should expect the policy of uniformly treating an individual to be worse as compared to the policy that assigns treatment to the top-k percentile of the input CATE estimates. Hence, the Qini score is qualitatively different than most of the other metrics as it does not directly compare the CATE estimates, rather it evaluates whether the groups most likely to be treated would actually benefit from the treatment. Since we only described the case of unbiased group ATE estimates computed using DR pseudo-outcomes (Cal DR Score, Qini DR Score), we now provide the same for the case of TMLE for convenience.

$$M^{\text{Cal-TMLE}}(\hat{\tau}) := \sum_{k=1}^{K} |G_k(\hat{\tau})| \left| \widehat{\text{GATE}}_k(\hat{\tau}) - \text{GATE}_k^{\text{TMLE}}(\hat{\tau}) \right| \qquad \text{(Cal TMLE Score)}$$

$$M^{\text{Qini-TMLE}}(\hat{\tau}) := \sum_{k=1}^{100} |G_{\geq k}(\hat{\tau})| \left( \text{GATE}_{\geq k}^{\text{TMLE}}(\hat{\tau}) - \text{ATE}^{\text{TMLE}} \right) \qquad \text{(Qini TMLE Score)}$$

where $\text{GATE}_k^{\text{TMLE}}(\hat{\tau}) := \frac{1}{|G_k(\hat{\tau})|} \sum_{i \in G_k} \tilde{\tau}_{\text{TMLE}}(x_i)$, $\quad \widehat{\text{GATE}}_k(\hat{\tau}) := \frac{1}{|G_k(\hat{\tau})|} \sum_{i \in G_k(\hat{\tau})} \hat{\tau}(x_i)$

$\text{GATE}_{\geq k}^{\text{TMLE}}(\hat{\tau}) := \frac{1}{|G_{\geq k}(\hat{\tau})|} \sum_{i \in G_{\geq k}} \tilde{\tau}_{\text{TMLE}}(x_i)$, $\quad \text{ATE}^{\text{TMLE}} := \frac{1}{N} \sum_{i=1}^{N} \tilde{\tau}_{\text{TMLE}}(x_i)$

## C   EXPERIMENT SETUP DETAILS

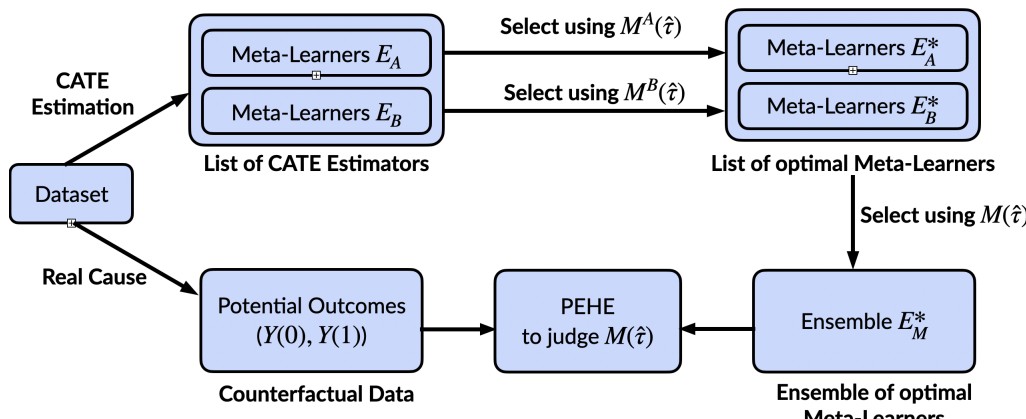

Figure 1: The proposed framework for comparing the different surrogate model selection strategies $M(\hat{\tau})$. We first perform intra-meta-learner selection using meta-learner based metrics, and then construct an ensemble over the optimal meta-learners using the input surrogate metric $M(\hat{\tau})$. Further, RealCause enables us to sample counterfactual data for realistic datases as well and benchmark the performance of each surrogate metric $M(\hat{\tau})$ as the PEHE of the ensemble returned by it.

### C.1   DATASET STATISTICS

| Dataset Group | ACIC 2016 | LaLonde CPS | LaLonde PSID | TWINS |
|---|---|---|---|---|
| Training Data Size | 3841 | 8089 | 1338 | 5992 |
| Evaluation Data Size | 961 | 6470 | 1069 | 4794 |
| Covariate Dimension | 82 | 8 | 8 | 75 |

Table 4: Statistics for the various datasets used in our analysis

| Dataset | LaLonde CPS | LaLonde PSID | TWINS |
|---|---|---|---|
| CATE (Mean) | $-10274.49$ | $-57280.61$ | $-0.02$ |
| CATE (Variance) | $3.73e+07$ | $5.92e+08$ | $0.01$ |
| Treatment Class % (Train) | $0.01$ | $0.06$ | $0.74$ |
| Treatment Class % (Eval) | $0.01$ | $0.06$ | $0.75$ |

Table 5: Extra statistics for the RealCause datasets used in our analysis. Train and Eval correspond to the different splits of the dataset used for training and evaluating CATE estimators.

We provide details regarding the training and evaluation sample size, along with the covariate dimensions in Table 4. We also provide more details regarding the mean and variance of CATE as well as the true propensity (fraction of samples in the treatment class) in Table 5 for the realistic datasets, and the same for ACIC 2016 datasets in Table 6. Note that the ACIC 2016 synthetic benchmark contains several datasets, where we had discarded datasets that have variance in true CATE lower than 0.01 to ensure heterogeneity; which leaves us with 75 datasets. Further, LaLonde CPS, LaLonde PSID and TWINS are the realistic benchmarks that do not contain the counterfactual potential outcomes for each individual. Hence, we used RealCause (Neal et al., 2020) to model the counterfactual potential outcomes for these datasets, essentially making them *semi-synthetic datasets*.

## C.2 CATE Estimators Implementation Details

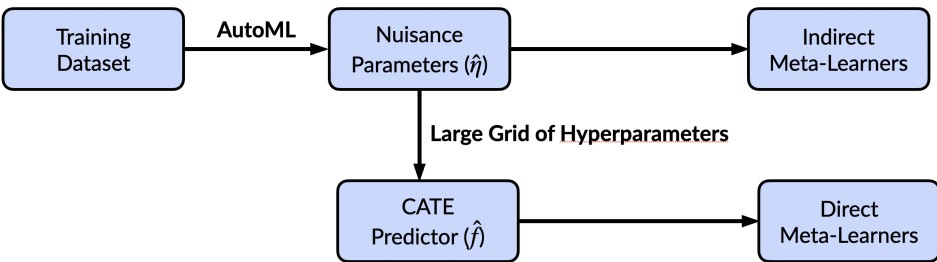

Figure 2: Illustrating the construction of indirect and direct meta-learners used in our empirical study. We use a large grid over different regression model classes and hyperparameters for choosing the CATE predictor in direct meta-learners, resulting in 103 different CATE estimators per direct meta-learner. This design choice is an improvement from prior works which consider only a few choices for hyperparameters of direct meta-learners.

For each dataset, we consider 7 different type of meta-learners for CATE estimation, including both indirect and direct meta-learners. For indirect meta-learners, we consider both the S-Learner and T-Learner; while for direct meta-learners, we consider the Projected S-Learner, X-Learner, DR-Learner, R-Learner, and Causal Forest Learner. Details regarding each of these meta-learners can be found in Appendix A. Note that we do not consider deep learning based CATE estimators since the scale of our study is already quite large. Also, the basic ML techniques with appropriate hyperparameter tuning should have good performance as well since we work with tabular datasets.

The nuisance models ($\hat{\eta}$) associated with all these various meta-learners are learned using AutoML (Wang et al., 2021) with a budget of 30 minutes. Specifically, we have five different types of nuisance models; propensity model ($\hat{\pi}(X)$), outcome model used in S-Learner, Projected S-Learner, DR-Learner ($\hat{\mu}(X, W)$), outcome models used in T-Learner, X-Learner ($\hat{\mu}_0(X)$, $\hat{\mu}_1(X)$), and outcome model for R-Learner, Causal Forest Learner ($\hat{\mu}(X)$); each learned using AutoML.

Further, for the CATE predictor ($\hat{f}$) in direct meta-learners, we use a total of 103 different regression models with variation across regression model class and associated hyperparameters, specified below. We used *sklearn* for implementing all the regression models and we use the same notation from sklearn for representing the regression model class and the corresponding hyperparameter names. For representing the hyperparameter ranges, we use the *numpy* logspace function to sample from a range (unless specified otherwise), with the syntax (start value, end value, total entries).

- Linear Regression; No Hyperparameter
- Linear Regression; Degree 2 polynomial features; No interaction terms; No Hypereparameter
- Linear Regression; Degree 2 polynomial features; Interaction term; No Hyperparameter
- Ridge Regression; Hyperparameters ($\alpha$): $np.logspace(-4, 5, 10)$
- Kernel Ridge Regression; Hyperparameters ($\alpha$): $np.logspace(-4, 5, 10)$
- Lasso Regression; Hyperparameters ($\alpha$): $np.logspace(-4, 5, 10)$
- Elastic Net Regression; Hyperparameters ($\alpha$): $np.logspace(-4, 5, 10)$
- SVR; Sigmoid Kernel; Hyperparameters ($C$): $np.logspace(-4, 5, 10)$
- SVR; RBF Kernel; Hyperparameters ($C$): $np.logspace(-4, 5, 10)$
- Linear SVR; Hyperparameters ($C$): $np.logspace(-4, 5, 10)$
- Decision Tree: Hyperparameters ($max\ depth$): $list(range(2, 11)) + [None]$
- Random Forest: Hyperparameters ($max\ depth$): $list(range(2, 11)) + [None]$
- Gradient Boosting: Hyperparameters ($max\ depth$): $list(range(2, 11)) + [None]$

Hence, we have 103 different CATE estimators for each type of direct meta-learner, and a single CATE estimator for each type of indirect meta-learner, resulting in a total of **415 CATE estimators** per dataset. This difference between the construction of indirect vs direct meta-learners is also visualized in Figure 2. Note that Causal Forest DML-Learner technically is a direct meta-learner, but we use the default random forest provided in the EconML implementation (Battocchi et al., 2019), hence effectively we do not consider variations across its CATE predictor ($\hat{f}$) function.

## C.3 SURROGATE METRICS IMPLEMENTATION DETAILS

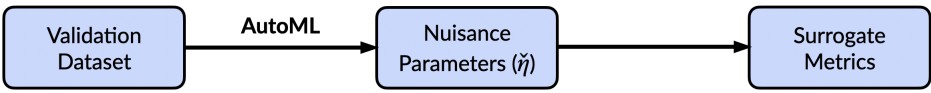

Figure 3: Illustrating the use of AutoML in selecting the nuisance parameters of surrogate metrics for CATE model selection. This is an important design choice in contrast to prior works which relied on small grid searches to infer the nuisance parameters associated with surrogate metrics, potentially resulting in biased estimates and affecting the model selection ability of surrogate metrics.

Please refer to Appendix B for full details regarding the implementation for each surrogate criteria of CATE model selection. An important point to highlight is that for surrogate metrics based on direct meta-learner strategies, we only consider the pseudo-outcome variant. E.g., DR Score only uses pseudo-outcomes to approximate $\tilde{\tau}(x)$ and does not train any regression model for direct CATE estimation (like we do in DR-Learner). Further, all nuisance models ($\check{\eta}$) are trained on the validation set using AutoML, specifically FLAML (Wang et al., 2021), with a budget of 30 minutes (Figure 3).

Regarding the *two-level model selection strategy*, each metric $M^J(\hat{\tau})$ for selecting amongst meta-estimators of type $J$ is provided below.

- DR T Score for selecting amongst DR-Learners
- R Score for selecting amongst DML-Learners
- X Score for selecting amongst X-Learners
- S Score for selecting amongst Projected S-Learners

Note that for the remaining meta-learners (S-Learner, T-Learner, Causal Forest Learner) we did not have any hyperparameters to select over as their nuisance models have been learned via AutoML. Hence, the first step in the two-level model selection strategy is trivial for them and they can be directly for the second step of model selection via a general surrogate metric.

| Dataset | CATE (Mean) | CATE (Variance) | Treatment Class % (Train) | Treatment Class % (Eval) |
|---|---|---|---|---|
| ACIC_2016_0 | 1.44 | 10.99 | 0.29 | 0.29 |
| ACIC_2016_3 | 2.99 | 29.36 | 0.37 | 0.35 |
| ACIC_2016_4 | 5.48 | 30.62 | 0.28 | 0.30 |
| ACIC_2016_5 | 3.38 | 4.46 | 0.29 | 0.29 |
| ACIC_2016_6 | 3.79 | 6.85 | 0.28 | 0.30 |
| ACIC_2016_7 | 4.45 | 9.40 | 0.30 | 0.30 |
| ACIC_2016_8 | 4.00 | 9.27 | 0.29 | 0.30 |
| ACIC_2016_9 | 2.06 | 8.44 | 0.32 | 0.31 |
| ACIC_2016_10 | 3.71 | 11.77 | 0.29 | 0.31 |
| ACIC_2016_11 | 5.32 | 26.08 | 0.32 | 0.30 |
| ACIC_2016_12 | 2.95 | 17.74 | 0.50 | 0.49 |
| ACIC_2016_13 | 3.27 | 7.14 | 0.50 | 0.54 |
| ACIC_2016_14 | 2.34 | 45.37 | 0.47 | 0.46 |
| ACIC_2016_15 | 4.48 | 46.08 | 0.57 | 0.55 |
| ACIC_2016_16 | 3.80 | 23.87 | 0.57 | 0.56 |
| ACIC_2016_17 | 0.24 | 18.76 | 0.37 | 0.38 |
| ACIC_2016_18 | 3.16 | 16.24 | 0.52 | 0.56 |
| ACIC_2016_19 | 1.85 | 77.99 | 0.52 | 0.51 |
| ACIC_2016_20 | 4.45 | 5.94 | 0.29 | 0.27 |
| ACIC_2016_21 | 6.39 | 56.62 | 0.32 | 0.32 |
| ACIC_2016_22 | 5.75 | 3.96 | 0.33 | 0.35 |
| ACIC_2016_23 | 3.44 | 11.47 | 0.27 | 0.26 |
| ACIC_2016_24 | 4.73 | 5.97 | 0.35 | 0.35 |
| ACIC_2016_25 | 3.92 | 6.93 | 0.31 | 0.32 |
| ACIC_2016_26 | 2.18 | 40.74 | 0.31 | 0.29 |
| ACIC_2016_27 | 3.19 | 23.88 | 0.33 | 0.31 |
| ACIC_2016_28 | 3.25 | 5.42 | 0.33 | 0.36 |
| ACIC_2016_29 | 3.75 | 9.41 | 0.35 | 0.36 |
| ACIC_2016_30 | 3.35 | 10.82 | 0.36 | 0.34 |
| ACIC_2016_31 | 3.84 | 20.17 | 0.35 | 0.34 |
| ACIC_2016_32 | 2.83 | 14.85 | 0.33 | 0.34 |
| ACIC_2016_33 | 5.22 | 19.58 | 0.63 | 0.63 |
| ACIC_2016_34 | 5.40 | 4.55 | 0.52 | 0.53 |
| ACIC_2016_35 | 2.72 | 22.20 | 0.36 | 0.35 |
| ACIC_2016_36 | 3.15 | 5.98 | 0.57 | 0.53 |
| ACIC_2016_37 | 5.38 | 11.32 | 0.60 | 0.63 |
| ACIC_2016_38 | 3.66 | 3.96 | 0.63 | 0.59 |
| ACIC_2016_39 | 5.12 | 4.23 | 0.63 | 0.63 |
| ACIC_2016_40 | 0.95 | 5.30 | 0.62 | 0.67 |
| ACIC_2016_41 | 3.87 | 3.66 | 0.59 | 0.57 |
| ACIC_2016_42 | 3.03 | 15.11 | 0.64 | 0.67 |
| ACIC_2016_43 | 3.87 | 25.95 | 0.66 | 0.65 |
| ACIC_2016_44 | 4.37 | 23.13 | 0.67 | 0.70 |
| ACIC_2016_45 | 4.93 | 25.60 | 0.65 | 0.69 |
| ACIC_2016_46 | 2.18 | 33.00 | 0.65 | 0.63 |
| ACIC_2016_47 | 5.35 | 1.21 | 0.32 | 0.35 |
| ACIC_2016_48 | 3.79 | 4.84 | 0.27 | 0.27 |
| ACIC_2016_49 | 5.85 | 1.06 | 0.23 | 0.21 |
| ACIC_2016_50 | 3.62 | 40.87 | 0.32 | 0.28 |
| ACIC_2016_51 | 4.15 | 12.54 | 0.19 | 0.18 |
| ACIC_2016_52 | 4.90 | 79.71 | 0.31 | 0.33 |
| ACIC_2016_53 | 2.58 | 33.89 | 0.29 | 0.29 |
| ACIC_2016_54 | 5.41 | 28.30 | 0.36 | 0.35 |
| ACIC_2016_55 | 3.63 | 4.37 | 0.36 | 0.36 |
| ACIC_2016_56 | 2.30 | 9.71 | 0.34 | 0.36 |
| ACIC_2016_57 | 3.31 | 13.43 | 0.36 | 0.34 |
| ACIC_2016_58 | 4.66 | 41.14 | 0.38 | 0.37 |
| ACIC_2016_59 | 4.84 | 40.29 | 0.38 | 0.35 |
| ACIC_2016_60 | 2.51 | 34.84 | 0.35 | 0.36 |
| ACIC_2016_61 | 4.23 | 30.48 | 0.35 | 0.34 |
| ACIC_2016_62 | 4.00 | 1.29 | 0.56 | 0.57 |
| ACIC_2016_63 | 3.31 | 15.72 | 0.51 | 0.49 |
| ACIC_2016_64 | 3.22 | 9.29 | 0.54 | 0.57 |
| ACIC_2016_65 | 3.90 | 28.30 | 0.56 | 0.55 |
| ACIC_2016_66 | 2.16 | 38.81 | 0.57 | 0.62 |
| ACIC_2016_67 | 5.41 | 117.43 | 0.61 | 0.60 |
| ACIC_2016_68 | 3.04 | 18.27 | 0.60 | 0.58 |
| ACIC_2016_69 | 2.82 | 4.75 | 0.65 | 0.63 |
| ACIC_2016_70 | 2.48 | 17.11 | 0.65 | 0.65 |
| ACIC_2016_71 | 2.55 | 18.66 | 0.69 | 0.65 |
| ACIC_2016_72 | 1.58 | 4.61 | 0.63 | 0.62 |
| ACIC_2016_73 | 4.31 | 5.44 | 0.63 | 0.62 |
| ACIC_2016_74 | 4.49 | 34.16 | 0.66 | 0.67 |
| ACIC_2016_75 | 3.20 | 29.85 | 0.64 | 0.63 |
| ACIC_2016_76 | 4.96 | 15.60 | 0.63 | 0.61 |

Table 6: Extra statistics for the various ACIC 2016 datasets used in our analysis. Train and Eval correspond to the different splits of the dataset used for training and evaluating CATE estimators.

# D   ADDITIONAL RESULTS

| Metric | ACIC 2016 | LaLonde CPS | LaLonde PSID | TWINS |
|---|---|---|---|---|
| Value Score | 1.05e+7 (4.31e+6) | 6.63 (5.52) | **0.48 (0.06)** | 0.57 (0.15) |
| Value DR Score | 13.02 (11.73) | 2.33 (1.41) | **0.46 (0.05)** | 1.61 (1.02) |
| Value DR Clip Score | 13.02 (11.73) | 0.43 (0.09) | 3.96 (3.51) | 1.68 (1.02) |
| Match Score | 3.60 (0.16) | **0.23 (0.04)** | 0.50 (0.06) | 0.38 (0.08) |
| S Score | 0.95 (0.02) | 0.90 (0.04) | 0.74 (0.04) | **0.29 (0.05)** |
| T Score | **0.56 (0.02)** | **0.16 (0.03)** | **0.42 (0.03)** | **0.31 (0.05)** |
| X Score | **0.56 (0.02)** | **0.16 (0.03)** | **0.41 (0.03)** | 0.35 (0.06) |
| R Score | 4.0 (0.11) | 0.83 (0.04) | 0.67 (0.03) | 0.60 (0.11) |
| Influence Score | 1455.75 (1439.46) | 0.95 (0.04) | 0.80 (0.02) | 1.08 (0.1) |
| Influence Clip Score | 1449.74 (1439.72) | 87.13 (60.29) | 1.16 (0.16) | 1.06 (0.09) |
| IPW Score | 3.21 (0.12) | **0.25 (0.05)** | **0.32 (0.02)** | 0.37 (0.06) |
| IPW Clip Score | 3.21 (0.12) | 0.60 (0.07) | **0.32 (0.02)** | 0.37 (0.06) |
| IPW Switch S Score | 3.21 (0.12) | **0.29 (0.06)** | **0.31 (0.02)** | 0.37 (0.06) |
| IPW Switch T Score | 3.21 (0.12) | **0.29 (0.06)** | **0.31 (0.02)** | 0.37 (0.06) |
| IPW CAB S Score | 3.20 (0.12) | **0.29 (0.06)** | **0.31 (0.02)** | 0.37 (0.06) |
| IPW CAB T Score | 3.20 (0.12) | **0.29 (0.06)** | **0.31 (0.02)** | 0.37 (0.06) |
| DR S Score | 0.93 (0.02) | 0.85 (0.05) | 0.73 (0.04) | 0.35 (0.06) |
| DR S Clip Score | 0.93 (0.02) | 0.90 (0.04) | 0.74 (0.04) | 0.34 (0.06) |
| DR T Score | **0.56 (0.02)** | **0.16 (0.02)** | **0.41 (0.03)** | **0.32 (0.07)** |
| DR T Clip Score | **0.56 (0.02)** | **0.16 (0.03)** | **0.41 (0.03)** | **0.33 (0.06)** |
| DR Switch S Score | 0.93 (0.02) | 0.90 (0.04) | 0.75 (0.04) | **0.32 (0.06)** |
| DR Switch T Score | **0.56 (0.02)** | **0.16 (0.03)** | **0.41 (0.03)** | **0.28 (0.05)** |
| DR CAB S Score | 0.93 (0.02) | 0.90 (0.04) | 0.74 (0.04) | 0.34 (0.06) |
| DR CAB T Score | **0.56 (0.02)** | **0.16 (0.03)** | **0.41 (0.03)** | **0.33   ( 0.06 )** |
| TMLE S Score | 1.06 (0.04) | 0.91 (0.04) | 0.74 (0.04) | **0.26 (0.05)** |
| TMLE T Score | **0.64 (0.03)** | **0.16 (0.03)** | 0.42 (0.03) | **0.31 (0.05)** |
| Cal DR S Score | 5.78 (0.19) | 0.87 (0.05) | 0.72 (0.04) | **0.19 (0.03)** |
| Cal DR T Score | 3.45 (0.11) | **0.17 (0.03)** | **0.42 (0.03)** | **0.21 (0.03)** |
| Cal TMLE S Score | 6.12 (0.21) | 0.90 (0.04) | 0.75 (0.04) | **0.17 (0.02)** |
| Cal TMLE T Score | 4.63 (0.18) | **0.17 (0.03)** | **0.42 (0.03)** | **0.22 (0.03)** |
| Qini DR S Score | 1.54 (0.06) | 917.0 (896.57) | 16.12 (15.38) | 1.78e+6 (1.77e+6) |
| Qini DR T Score | 1.32 (0.07) | 2.87 (1.53) | 0.57 (0.05) | 2.08e+7 (1.90e+7) |
| Qini TMLE S Score | 1.55 (0.06) | 295.06 (173.91) | 0.70 (0.04) | 1.78e+6 (1.78e+6) |
| Qini TMLE T Score | 1.32 (0.07) | 2.59 (1.49) | 0.57 (0.04) | 1.78e+6 (1.78e+6) |

Table 7: Normalized PEHE of the **best estimators** chosen by each metric with the **single-level model selection strategy**; results report the mean (standard error) across 20 seeds and also across datasets for the ACIC 2016 benchmark. **Lower value is better.**

| Metric | ACIC 2016 | LaLonde CPS | LaLonde PSID | TWINS |
|---|---|---|---|---|
| Value Score | 3.97 (1.98) | **0.34 (0.09)** | **0.43 (0.03)** | **0.21 (0.03)** |
| Value DR Score | **0.64 (0.03)** | **0.25 (0.08)** | **0.47 (0.04)** | **0.21 (0.03)** |
| Value DR Clip Score | **0.64 (0.03)** | **0.22 (0.06)** | **0.47 (0.04)** | **0.21 (0.03)** |
| Match Score | 1.76 (0.09) | **0.17 (0.03)** | **0.45 (0.03)** | **0.21 (0.03)** |
| S Score | 0.93 (0.02) | 0.90 (0.04) | 0.75 (0.04) | **0.21 (0.03)** |
| T Score | **0.56 (0.02)** | **0.16 (0.03)** | **0.41 (0.03)** | **0.21 (0.03)** |
| X Score | **0.56 (0.02)** | **0.16 (0.03)** | **0.41 (0.03)** | **0.21 (0.03)** |
| R Score | 3.88 (0.11) | 0.86 (0.03) | 0.62 (0.03) | **0.21 (0.03)** |
| Influence Score | 3.26 (0.1) | 0.93 (0.04) | 0.77 (0.03) | **0.16 (0.02)** |
| Influence Clip Score | 3.19 (0.1) | 68.67 (27.37) | 1.26 (0.23) | **0.16 (0.02)** |
| IPW Score | 1.41 (0.06) | **0.16 (0.04)** | **0.38 (0.02)** | **0.21 (0.03)** |
| IPW Clip Score | 1.40 (0.06) | **0.18 (0.04)** | **0.37 (0.02)** | **0.21 (0.03)** |
| IPW Switch S Score | 1.41 (0.06) | **0.16 (0.04)** | **0.37 (0.02)** | **0.21 (0.03)** |
| IPW Switch T Score | 1.41 (0.06) | **0.16 (0.04)** | **0.37 (0.02)** | **0.21 (0.03)** |
| IPW CAB S Score | 1.40 (0.06) | **0.16 (0.04)** | **0.38 (0.02)** | **0.21 (0.03)** |
| IPW CAB T Score | 1.40 (0.06) | **0.16 (0.04)** | **0.38 (0.02)** | **0.21 (0.03)** |
| DR S Score | 0.92 (0.02) | 0.85 (0.04) | 0.74 (0.04) | **0.20 (0.03)** |
| DR S Clip Score | 0.92 (0.02) | 0.90 (0.04) | 0.74 (0.04) | **0.20 (0.03)** |
| DR T Score | **0.56 (0.02)** | **0.16 (0.02)** | **0.41 (0.03)** | **0.21 (0.03)** |
| DR T Clip Score | **0.56 (0.02)** | **0.14 (0.03)** | **0.41 (0.03)** | **0.21 (0.03)** |
| DR Switch S Score | 0.92 (0.02) | 0.90 (0.04) | 0.74 (0.04) | **0.21 (0.03)** |
| DR Switch T Score | **0.56 (0.02)** | **0.16 (0.03)** | **0.41 (0.03)** | **0.21 (0.03)** |
| DR CAB S Score | 0.92 (0.02) | 0.90 (0.04) | 0.74 (0.04) | **0.20 (0.03)** |
| DR CAB T Score | **0.56 (0.02)** | **0.16 (0.03)** | **0.41 (0.03)** | **0.21 (0.03)** |
| TMLE S Score | 1.0 (0.03) | 0.90 (0.04) | 0.75 (0.04) | **0.21 (0.03)** |
| TMLE T Score | **0.61 (0.03)** | **0.16 (0.03)** | **0.42 (0.03)** | **0.21 (0.03)** |
| Cal DR S Score | 1.18 (0.03) | 0.86 (0.04) | 0.74 (0.04) | **0.21 (0.03)** |
| Cal DR T Score | **0.62 (0.02)** | **0.19 (0.04)** | **0.42 (0.03)** | **0.22 (0.03)** |
| Cal TMLE S Score | 1.39 (0.05) | 0.90 (0.04) | 0.74 (0.04) | **0.18 (0.02)** |
| Cal TMLE T Score | 0.82 (0.02) | **0.2 (0.04)** | **0.42 (0.03)** | **0.21 (0.02)** |
| Qini DR S Score | **0.62 (0.02)** | 21.18 (15.26) | 0.68 (0.12) | **0.21 (0.03)** |
| Qini DR T Score | **0.58 (0.02)** | **0.14 (0.03)** | 0.52 (0.03) | **0.24 (0.04)** |
| Qini TMLE S Score | 0.67 (0.03) | 14.68 (9.97) | 0.60 (0.07) | **0.21 (0.03)** |
| Qini TMLE T Score | **0.63 (0.03)** | **0.15 (0.03)** | 0.53 (0.03) | **0.21 (0.03)** |

Table 8: Normalized PEHE of the **best estimators** chosen by each metric with the **two-level model selection strategy**; results report the mean (standard error) across 20 seeds and also across datasets for the ACIC 2016 benchmark. **Lower value is better.**

| Metric | ACIC 2016 | LaLonde CPS | LaLonde PSID | TWINS |
|---|---|---|---|---|
| Value Score | 1.94 (0.31) | 0.58 (0.08) | 0.47 (0.03) | **0.21** (**0.03**) |
| Value DR Score | 0.71 (0.03) | 0.37 (0.08) | 0.51 (0.04) | **0.21** (**0.03**) |
| Value DR Clip Score | 0.71 (0.03) | **0.27** (**0.07**) | 0.51 (0.04) | **0.21** (**0.03**) |
| Match Score | 1.56 (0.07) | **0.17** (**0.03**) | **0.45** (**0.03**) | **0.21** (**0.03**) |
| S Score | 0.86 (0.02) | 0.90 (0.04) | 0.75 (0.04) | **0.21** (**0.03**) |
| T Score | **0.45** (**0.02**) | **0.16** (**0.03**) | **0.41** (**0.03**) | **0.21** (**0.03**) |
| X Score | **0.45** (**0.02**) | **0.16** (**0.03**) | **0.41** (**0.03**) | **0.21** (**0.03**) |
| R Score | 3.33 (0.09) | 0.86 (0.03) | 0.62 (0.03) | **0.21** (**0.03**) |
| Influence Score | 2.98 (0.11) | 0.93 (0.04) | 0.77 (0.03) | **0.16** (**0.02**) |
| Influence Clip Score | 2.89 (0.11) | 68.67 (27.37) | 1.26 (0.23) | **0.16** (**0.02**) |
| IPW Score | 1.33 (0.05) | **0.16** (**0.04**) | **0.38** (**0.02**) | **0.20** (**0.03**) |
| IPW Clip Score | 1.32 (0.05) | **0.18** (**0.04**) | **0.37** (**0.02**) | **0.20** (**0.03**) |
| IPW Switch S Score | 1.33 (0.05) | **0.16** (**0.04**) | **0.37** (**0.02**) | **0.20** (**0.03**) |
| IPW Switch T Score | 1.33 (0.05) | **0.16** (**0.04**) | **0.37** (**0.02**) | **0.20** (**0.03**) |
| IPW CAB S Score | 1.33 (0.06) | **0.16** (**0.04**) | **0.38** (**0.02**) | **0.20** (**0.03**) |
| IPW CAB T Score | 1.33 (0.06) | **0.16** (**0.04**) | **0.38** (**0.02**) | **0.20** (**0.03**) |
| DR S Score | 0.85 (0.02) | 0.85 (0.04) | 0.74 (0.04) | **0.20** (**0.03**) |
| DR S Clip Score | 0.85 (0.02) | 0.90 (0.04) | 0.74 (0.04) | **0.21** (**0.03**) |
| DR T Score | **0.45** (**0.02**) | **0.16** (**0.02**) | **0.41** (**0.03**) | **0.21** (**0.03**) |
| DR T Clip Score | **0.45** (**0.02**) | **0.14** (**0.03**) | **0.41** (**0.03**) | **0.21** (**0.03**) |
| DR Switch S Score | 0.85 (0.02) | 0.90 (0.04) | 0.74 (0.04) | **0.21** (**0.03**) |
| DR Switch T Score | **0.45** (**0.02**) | **0.16** (**0.03**) | **0.41** (**0.03**) | **0.21** (**0.03**) |
| DR CAB S Score | 0.85 (0.02) | 0.90 (0.04) | 0.74 (0.04) | **0.21** (**0.03**) |
| DR CAB T Score | **0.45** (**0.02**) | **0.16** (**0.03**) | **0.41** (**0.03**) | **0.21** (**0.03**) |
| TMLE S Score | 0.93 (0.03) | 0.90 (0.04) | 0.75 (0.04) | **0.21** (**0.03**) |
| TMLE T Score | **0.51** (**0.03**) | **0.16** (**0.03**) | **0.42** (**0.03**) | **0.21** (**0.03**) |
| Cal DR S Score | 1.42 (0.25) | 0.89 (0.04) | 0.74 (0.04) | **0.20** (**0.03**) |
| Cal DR T Score | 0.62 (0.02) | **0.20** (**0.04**) | **0.42** (**0.03**) | **0.20** (**0.03**) |
| Cal TMLE S Score | 1.43 (0.25) | 0.89 (0.04) | 0.74 (0.04) | **0.21** (**0.03**) |
| Cal TMLE T Score | 0.82 (0.02) | **0.19** (**0.04**) | **0.42** (**0.03**) | **0.20** (**0.03**) |
| Qini DR S Score | 0.62 (0.02) | 24.86 (14.92) | 0.63 (0.04) | **0.21** (**0.03**) |
| Qini DR T Score | 0.58 (0.02) | **0.14** (**0.03**) | 0.50 (0.03) | **0.21** (**0.03**) |
| Qini TMLE S Score | 0.67 (0.03) | 11.96 (4.04) | 0.63 (0.04) | **0.21** (**0.03**) |
| Qini TMLE T Score | 0.63 (0.03) | **0.15** (**0.03**) | 0.51 (0.02) | **0.21** (**0.03**) |

Table 9: Normalized PEHE of the **ensemble estimators** chosen by each metric with the **two-level model selection strategy**; results report the mean (standard error) across 20 seeds and also across datasets for the ACIC 2016 benchmark. **Lower value is better.**

| Metric | ACIC 2016 | LaLonde CPS | LaLonde PSID | TWINS |
|---|---|---|---|---|
| Value Score | 1.21$e$+7 (6.21$e$+6) | 1.15 (0.4) | 0.64 (0.09) | 0.57 (0.15) |
| Value DR Score | 13.11 (11.72) | 1.38 (0.41) | 0.61 (0.04) | 0.58 (0.14) |
| Value DR Clip Score | 13.11 (11.72) | 1.63 (0.46) | 0.59 (0.05) | 0.64 (0.16) |
| Match Score | 3.49 (0.16) | **0.43 (0.06)** | **0.51 (0.05)** | **0.38 (0.08)** |
| S Score | 0.98 (0.03) | 0.83 (0.05) | 0.75 (0.04) | **0.29 (0.05)** |
| T Score | **0.61 (0.02)** | **0.35 (0.05)** | **0.48 (0.04)** | **0.31 (0.05)** |
| X Score | **0.61 (0.02)** | **0.35 (0.05)** | **0.48 (0.04)** | **0.35 (0.06)** |
| R Score | 3.31 (0.08) | 0.83 (0.05) | 0.68 (0.03) | 0.60 (0.11) |
| Influence Score | 1.04$e$+6 (7.76$e$+5) | 0.91 (0.04) | 0.79 (0.03) | **0.19 (0.03)** |
| Influence Clip Score | 5.90$e$+5 (5.90$e$+5) | 2.18 (0.38) | 0.94 (0.06) | **0.2 (0.03)** |
| IPW Score | 2.53 (0.09) | **0.59 (0.1)** | **0.43 (0.03)** | 0.40 (0.07) |
| IPW Clip Score | 2.53 (0.09) | 0.80 (0.14) | **0.38 (0.04)** | 0.40 (0.07) |
| IPW Switch S Score | 2.53 (0.09) | **0.59 (0.1)** | **0.43 (0.03)** | 0.40 (0.07) |
| IPW Switch T Score | 2.53 (0.09) | **0.59 (0.1)** | **0.43 (0.03)** | 0.40 (0.07) |
| IPW CAB S Score | 2.53 (0.09) | **0.59 (0.1)** | **0.43 (0.03)** | 0.40 (0.07) |
| IPW CAB T Score | 2.53 (0.09) | **0.59 (0.1)** | **0.43 (0.03)** | 0.40 (0.07) |
| DR S Score | 0.95 (0.02) | 0.81 (0.06) | 0.74 (0.04) | 0.37 (0.06) |
| DR S Clip Score | 0.95 (0.02) | 0.83 (0.05) | 0.75 (0.04) | **0.35 (0.06)** |
| DR T Score | **0.61 (0.02)** | **0.35 (0.05)** | **0.48 (0.04)** | 0.32 (0.07) |
| DR T Clip Score | **0.61 (0.02)** | **0.35 (0.05)** | **0.48 (0.04)** | 0.33 (0.06) |
| DR Switch S Score | 0.95 (0.02) | 0.83 (0.05) | 0.75 (0.04) | **0.32 (0.06)** |
| DR Switch T Score | **0.61 (0.02)** | **0.35 (0.05)** | **0.48 (0.04)** | **0.28 (0.05)** |
| DR CAB S Score | 0.95 (0.02) | 0.83 (0.05) | 0.75 (0.04) | **0.35 (0.06)** |
| DR CAB T Score | **0.61 (0.02)** | **0.35 (0.05)** | **0.48 (0.04)** | 0.33 (0.06) |
| TMLE S Score | 1.04 (0.03) | 0.83 (0.05) | 0.76 (0.04) | **0.26 (0.05)** |
| TMLE T Score | **0.65 (0.02)** | **0.35 (0.05)** | **0.49 (0.03)** | **0.31 (0.05)** |
| Cal DR S Score | 5.77 (0.14) | 0.87 (0.07) | 0.74 (0.04) | **0.23 (0.04)** |
| Cal DR T Score | 3.60 (0.1) | **0.38 (0.05)** | **0.48 (0.03)** | **0.22 (0.04)** |
| Cal TMLE S Score | 6.09 (0.19) | 0.85 (0.05) | 0.75 (0.04) | **0.23 (0.05)** |
| Cal TMLE T Score | 4.62 (0.15) | **0.39 (0.04)** | **0.49 (0.03)** | **0.24 (0.03)** |
| Qini DR S Score | 1.36 (0.05) | 8.89 (2.74) | 0.68 (0.08) | 1.78$e$+6 (1.77$e$+6) |
| Qini DR T Score | 1.16 (0.05) | 0.79 (0.08) | 0.66 (0.05) | 1.78$e$+6 (1.77$e$+6) |
| Qini TMLE S Score | 1.41 (0.06) | 21.78 (14.52) | 1.39 (0.67) | 1.78$e$+6 (1.77$e$+6) |
| Qini TMLE T Score | 1.17 (0.04) | 0.89 (0.12) | 0.65 (0.05) | 1.78$e$+6 (1.77$e$+6) |

Table 10: Normalized PEHE of the **best estimators** chosen by each metric amongst only the set of **DR-Learners**; results report the mean (standard error) across 20 seeds and also across datasets for the ACIC 2016 benchmark. **Lower value is better.**

| Metric | ACIC 2016 | LaLonde CPS | LaLonde PSID | TWINS |
|---|---|---|---|---|
| Value Score | 2.78e+6 (1.58e+6) | 800.39 (792.56) | **0.46** (**0.05**) | 1.21e+4 (1.21e+4) |
| Value DR Score | 7.63e+5 (3.83e+5) | 3.78 (1.82) | 21.25 (15.63) | 1.21e+4 (1.21e+4) |
| Value DR Clip Score | 7.63e+5 (3.83e+5) | 2.22 (0.77) | 21.25 (15.63) | 1.21e+4 (1.21e+4) |
| Match Score | 5.48 (0.17) | **0.27** (**0.05**) | **0.48** (**0.06**) | **0.17** (**0.04**) |
| S Score | **4.39** (**0.14**) | 0.9 (0.04) | 0.73 (0.03) | **0.10** (**0.03**) |
| T Score | **4.26** (**0.14**) | **0.2** (**0.03**) | **0.43** (**0.03**) | **0.12** (**0.03**) |
| X Score | **4.26** (**0.14**) | **0.2** (**0.03**) | **0.43** (**0.03**) | **0.13** (**0.03**) |
| R Score | 6.35 (0.15) | 0.86 (0.03) | 0.64 (0.04) | **0.32** (**0.1**) |
| Influence Score | 2.81e+4 (2.49e+4) | 0.96 (0.03) | 0.81 (0.03) | 1.08 (0.1) |
| Influence Clip Score | 1.23e+4 (1.23e+4) | 58.5 (56.51) | 1.24 (0.17) | 1.06 (0.09) |
| IPW Score | 5.87 (0.18) | 0.38 (0.06) | **0.33** (**0.02**) | **0.14** (**0.03**) |
| IPW Clip Score | 5.87 (0.19) | 0.57 (0.07) | **0.33** (**0.02**) | **0.12** (**0.03**) |
| IPW Switch S Score | 5.86 (0.19) | 0.41 (0.06) | **0.32** (**0.02**) | **0.12** (**0.03**) |
| IPW Switch T Score | 5.86 (0.19) | 0.41 (0.06) | **0.32** (**0.02**) | **0.12** (**0.03**) |
| IPW CAB S Score | 5.86 (0.18) | 0.41 (0.06) | **0.32** (**0.02**) | **0.12** (**0.03**) |
| IPW CAB T Score | 5.86 (0.18) | 0.41 (0.06) | **0.32** (**0.02**) | **0.12** (**0.03**) |
| DR S Score | **4.38** (**0.14**) | 0.84 (0.05) | 0.72 (0.03) | **0.13** (**0.04**) |
| DR S Clip Score | **4.38** (**0.14**) | 0.9 (0.04) | 0.72 (0.03) | **0.15** (**0.04**) |
| DR T Score | **4.26** (**0.14**) | **0.2** (**0.03**) | **0.43** (**0.03**) | **0.14** (**0.04**) |
| DR T Clip Score | **4.26** (**0.14**) | **0.2** (**0.03**) | **0.43** (**0.03**) | **0.14** (**0.04**) |
| DR Switch S Score | **4.38** (**0.14**) | 0.9 (0.04) | 0.73 (0.03) | **0.14** (**0.04**) |
| DR Switch T Score | **4.26** (**0.14**) | **0.2** (**0.03**) | **0.43** (**0.03**) | **0.15** (**0.04**) |
| DR CAB S Score | **4.38** (**0.14**) | 0.9 (0.04) | 0.72 (0.03) | **0.15** (**0.04**) |
| DR CAB T Score | **4.26** (**0.14**) | **0.2** (**0.03**) | **0.43** (**0.03**) | **0.14** (**0.04**) |
| TMLE S Score | **4.45** (**0.14**) | 0.91 (0.04) | 0.74 (0.03) | **0.1** (**0.03**) |
| TMLE T Score | **4.3** (**0.14**) | **0.18** (**0.03**) | **0.43** (**0.03**) | **0.13** (**0.03**) |
| Cal DR S Score | 6.23 (0.19) | 0.86 (0.05) | 0.72 (0.03) | **0.17** (**0.03**) |
| Cal DR T Score | 5.94 (0.18) | **0.19** (**0.03**) | **0.43** (**0.03**) | **0.19** (**0.03**) |
| Cal TMLE S Score | 6.28 (0.2) | 0.9 (0.04) | 0.74 (0.03) | **0.16** (**0.02**) |
| Cal TMLE T Score | 6.02 (0.2) | **0.19** (**0.03**) | **0.43** (**0.03**) | **0.19** (**0.02**) |
| Qini DR S Score | 6.19e+6 (3.75e+6) | 2.82e+3 (1.51e+3) | 19.4 (15.6) | 1.13e+7 (1.12e+7) |
| Qini DR T Score | 8.22e+6 (4.09e+6) | 3.47 (1.54) | 3.92 (3.51) | 1.13e+7 (1.13e+7) |
| Qini TMLE S Score | 7.38e+6 (4.08e+6) | 916.03 (599.74) | 6.34 (4.07) | 1.13e+7 (1.13e+7) |
| Qini TMLE T Score | 6.91e+6 (4.00e+6) | 4.71 (1.86) | 3.94 (3.51) | 2.13e+7 (1.47e+7) |

Table 11: Normalized PEHE of the **best estimators** chosen by each metric amongst only the set of **DML-Learners**; results report the mean (standard error) across 20 seeds and also across datasets for the ACIC 2016 benchmark. **Lower value is better.**

| Metric | ACIC 2016 | LaLonde CPS | LaLonde PSID | TWINS |
|---|---|---|---|---|
| Value Score | 1.04$e$+7 (4.23$e$+6) | 1.47 (0.53) | **0.45 (0.04)** | **0.28 (0.05)** |
| Value DR Score | 1.04 (0.1) | **0.35 (0.1)** | **0.42 (0.04)** | 0.29 (0.04) |
| Value DR Clip Score | 1.04 (0.1) | **0.39 (0.09)** | **0.38 (0.04)** | 0.29 (0.04) |
| Match Score | 3.40 (0.16) | **0.31 (0.05)** | **0.47 (0.05)** | **0.26 (0.04)** |
| S Score | 0.88 (0.02) | 0.72 (0.03) | 0.72 (0.04) | **0.22 (0.04)** |
| T Score | **0.34 (0.01)** | **0.20 (0.03)** | **0.42 (0.02)** | 0.31 (0.04) |
| X Score | **0.34 (0.01)** | **0.20 (0.03)** | **0.40 (0.02)** | 0.29 (0.04) |
| R Score | 3.14 (0.09) | 0.72 (0.03) | 0.61 (0.03) | **0.28 (0.05)** |
| Influence Score | 1.45$e$+3 (1.44$e$+3) | 0.75 (0.03) | 0.76 (0.02) | 0.41 (0.07) |
| Influence Clip Score | 1.45$e$+3 (1.44$e$+3) | 31.67 (24.68) | 1.32 (0.21) | 0.46 (0.09) |
| IPW Score | 2.40 (0.11) | 0.40 (0.06) | **0.33 (0.02)** | 0.29 (0.04) |
| IPW Clip Score | 2.39 (0.11) | 0.58 (0.08) | **0.33 (0.02)** | 0.29 (0.04) |
| IPW Switch S Score | 2.40 (0.11) | 0.40 (0.06) | **0.32 (0.02)** | 0.29 (0.04) |
| IPW Switch T Score | 2.40 (0.11) | 0.40 (0.06) | **0.32 (0.02)** | 0.29 (0.04) |
| IPW CAB S Score | 2.40 (0.11) | 0.40 (0.06) | **0.32 (0.02)** | 0.29 (0.04) |
| IPW CAB T Score | 2.40 (0.11) | 0.40 (0.06) | **0.32 (0.02)** | 0.29 (0.04) |
| DR S Score | 0.85 (0.02) | 0.68 (0.04) | 0.7 (0.05) | **0.28 (0.04)** |
| DR S Clip Score | 0.85 (0.02) | 0.71 (0.04) | 0.71 (0.04) | **0.28 (0.04)** |
| DR T Score | **0.34 (0.01)** | **0.20 (0.03)** | **0.40 (0.02)** | **0.28 (0.04)** |
| DR T Clip Score | **0.34 (0.01)** | **0.20 (0.03)** | **0.40 (0.02)** | **0.28 (0.04)** |
| DR Switch S Score | 0.85 (0.02) | 0.72 (0.03) | 0.71 (0.04) | **0.28 (0.04)** |
| DR Switch T Score | **0.34 (0.01)** | **0.20 (0.03)** | **0.40 (0.02)** | 0.32 (0.04) |
| DR CAB S Score | 0.85 (0.02) | 0.71 (0.04) | 0.71 (0.04) | **0.28 (0.04)** |
| DR CAB T Score | **0.34 (0.01)** | **0.20 (0.03)** | **0.40 (0.02)** | **0.28 (0.04)** |
| TMLE S Score | 0.96 (0.03) | 0.73 (0.03) | 0.73 (0.04) | **0.25 (0.04)** |
| TMLE T Score | **0.39 (0.02)** | **0.20 (0.03)** | **0.42 ( 0.02 )** | 0.31 (0.04) |
| Cal DR S Score | 3.15 (0.1) | 0.71 (0.04) | 0.71 (0.04) | **0.22 (0.03)** |
| Cal DR T Score | 1.40 (0.06) | **0.24 (0.04)** | **0.41 ( 0.03 )** | **0.22 (0.03)** |
| Cal TMLE S Score | 3.80 (0.11) | 0.79 (0.04) | 0.73 (0.04) | **0.17 (0.02)** |
| Cal TMLE T Score | 2.04 (0.07) | **0.2 (0.03)** | **0.41 ( 0.03 )** | **0.20 (0.03)** |
| Qini DR S Score | 0.59 (0.03) | 1.43 (0.75) | 0.62 (0.15) | 6.23$e$+5 (6.22$e$+5) |
| Qini DR T Score | 0.47 (0.03) | 0.89 (0.35) | **0.33 (0.04)** | 1.96$e$+7 (1.90$e$+7) |
| Qini TMLE S Score | 0.75 (0.04) | 2.89 (1.14) | 1.13 (0.6) | 6.23$e$+5 (6.22$e$+5) |
| Qini TMLE T Score | 0.51 (0.02) | 1.47 (0.44) | **0.31 (0.04)** | 6.23$e$+5 (6.22$e$+5) |

Table 12: Normalized PEHE of the **best estimators** chosen by each metric amongst only the set of **X-Learners**; results report the mean (standard error) across 20 seeds and also across datasets for the ACIC 2016 benchmark. **Lower value is better.**

| Metric | ACIC 2016 | LaLonde CPS | LaLonde PSID | TWINS |
|---|---|---|---|---|
| Value Score | 1.25$e$+7 (4.13$e$+6) | **0.83 (0.06)** | **0.73 (0.03)** | 2.39$e$+6 (2.39$e$+6) |
| Value DR Score | 1.24 (0.09) | **0.83 (0.05)** | **0.73 (0.03)** | **0.22 (0.04)** |
| Value DR Clip Score | 1.23 (0.09) | **0.83 (0.05)** | 8.53 (7.8) | **0.22 (0.04)** |
| Match Score | 3.36 (0.16) | **0.75 (0.06)** | **0.7 (0.02)** | 0.23 (0.04) |
| S Score | 0.9 (0.02) | **0.91 (0.05)** | **0.76 (0.04)** | **0.22 (0.04)** |
| T Score | **0.55 (0.01)** | **0.75 (0.06)** | **0.68 (0.02)** | **0.22 (0.04)** |
| X Score | **0.55 (0.01)** | **0.75 (0.06)** | **0.68 (0.02)** | **0.22 (0.04)** |
| R Score | 3.19 (0.09) | **0.88 (0.04)** | **0.72 (0.03)** | **0.22 (0.04)** |
| Influence Score | 1.32$e$+6 (9.54$e$+5) | **0.91 (0.04)** | 0.78 (0.02) | **0.17 (0.03)** |
| Influence Clip Score | 7.49$e$+5 (7.49$e$+5) | **0.94 (0.07)** | **0.83 (0.08)** | **0.16 (0.03)** |
| IPW Score | 2.33 (0.11) | **0.75 (0.06)** | **0.68 (0.02)** | 0.21 (0.04) |
| IPW Clip Score | 2.33 (0.11) | **0.75 (0.06)** | **0.68 (0.02)** | **0.22 (0.04)** |
| IPW Switch S Score | 2.33 (0.11) | **0.75 (0.06)** | **0.68 (0.02)** | 0.21 (0.04) |
| IPW Switch T Score | 2.33 (0.11) | **0.75 (0.06)** | **0.68 (0.02)** | 0.21 (0.04) |
| IPW CAB S Score | 2.33 (0.11) | **0.75 (0.06)** | **0.68 (0.02)** | 0.21 (0.04) |
| IPW CAB T Score | 2.33 (0.11) | **0.75 (0.06)** | **0.68 (0.02)** | 0.21 (0.04) |
| DR S Score | 0.88 (0.02) | **0.89 (0.04)** | **0.75 (0.04)** | **0.22 (0.04)** |
| DR S Clip Score | 0.88 (0.02) | **0.91 (0.04)** | **0.75 (0.04)** | 0.21 (0.04) |
| DR T Score | **0.55 (0.01)** | **0.75 (0.06)** | **0.68 (0.02)** | **0.22 (0.04)** |
| DR T Clip Score | **0.55 (0.01)** | **0.75 (0.06)** | **0.68 (0.02)** | **0.22 (0.04)** |
| DR Switch S Score | 0.88 (0.02) | **0.91 (0.04)** | **0.75 (0.04)** | **0.22 (0.04)** |
| DR Switch T Score | **0.55 (0.01)** | **0.75 (0.06)** | **0.68 (0.02)** | **0.22 (0.04)** |
| DR CAB S Score | 0.88 (0.02) | **0.91 (0.04)** | **0.75 (0.04)** | 0.21 (0.04) |
| DR CAB T Score | **0.55 (0.01)** | **0.75 (0.06)** | **0.68 (0.02)** | **0.22 (0.04)** |
| TMLE S Score | 0.95 (0.03) | **0.91 (0.04)** | **0.76 (0.04)** | **0.22 (0.04)** |
| TMLE T Score | **0.58 (0.01)** | **0.75 (0.06)** | **0.68 (0.02)** | **0.22 (0.04)** |
| Cal DR S Score | 5.14 (0.15) | **0.9 (0.05)** | **0.75 (0.04)** | **0.22 (0.03)** |
| Cal DR T Score | 3.88 (0.1) | **0.76 (0.06)** | **0.69 (0.02)** | **0.22 (0.03)** |
| Cal TMLE S Score | 5.66 (0.17) | **0.9 (0.05)** | **0.76 (0.04)** | **0.2 (0.03)** |
| Cal TMLE T Score | 4.51 (0.15) | **0.75 (0.06)** | **0.69 (0.02)** | 0.23 (0.03) |
| Qini DR S Score | 1.1 (0.03) | **0.85 (0.06)** | **0.8 (0.07)** | **0.22 (0.04)** |
| Qini DR T Score | 0.98 (0.03) | **0.81 (0.06)** | **0.75 (0.03)** | **0.2 (0.04)** |
| Qini TMLE S Score | 1.16 (0.05) | 185.85 (184.98) | **0.81 (0.07)** | **0.22 (0.04)** |
| Qini TMLE T Score | 0.98 (0.03) | **0.81 (0.06)** | **0.75 (0.03)** | **0.22 (0.04)** |

Table 13: Normalized PEHE of the **best estimators** chosen by each metric amongst only the set of **Projected S-Learners**; results report the mean (standard error) across 20 seeds and also across datasets for the ACIC 2016 benchmark. **Lower value is better.**

