# OpenReview forum: "Empirical Analysis of Model Selection for Heterogeneous Causal Effect Estimation"
_ICLR.cc/2024/Conference — ICLR 2024 spotlight_

### Official Review · Reviewer_3TcV · 2023-10-31

**Soundness:** 3 good
**Presentation:** 2 fair
**Contribution:** 2 fair
**Rating:** 5
**Confidence:** 3

**Summary:**

This work uses extensive empirical analysis to explore the possibility as well as the best way to do model selection for CATE estimation. This is a really challenging problem because we don't have an ideal model selection metric for this task. If we had access to hypothetical counterfactual data, then we could have used the ideal metric PEHE. Hence, the main difficulty stems from not observing both potential
outcomes for each sample, and we need to design model selection metrics which do not rely on counterfactual data.

However, this is an important direction to make causal inference more democratization, especially for those who are not familiar with statistics or causal inference.

**Strengths:**

This work uses extensive empirical analysis to explore the possibility as well as the best way to do model selection for CATE estimation. This is a really challenging problem because we don't have an ideal model selection metric for this task. If we had access to hypothetical counterfactual data, then we could have used the ideal metric PEHE. Hence, the main difficulty stems from not observing both potential
outcomes for each sample, and we need to design model selection metrics which do not rely on counterfactual data.

However, this is an important direction to make causal inference more democratization, especially for those who are not familiar with statistics or causal inference.

**Weaknesses:**

(1) Due to this is an empirical study, there may be some concerns about the results of the experiments, such as whether all experimental conditions have been taken into account, whether the way the training and validation datasets are partitioned will affect the final experimental conclusions, and etc.
(2) The general framework proposed in the paper seems to have high requirements for the amount of data.
(3) Some relevant papers were missing, e.g. Nguyen et al., OpportunityFinder: A Framework for Automated Causal Inference.

**Questions:**

(1) Due to this is an empirical study, there may be some concerns about the results of the experiments, such as whether all experimental conditions have been taken into account, whether the way the training and validation datasets are partitioned will affect the final experimental conclusions, and etc.
(2) The general framework proposed in the paper seems to have high requirements for the amount of data.
(3) Some relevant papers were missing, e.g. Nguyen et al., OpportunityFinder: A Framework for Automated Causal Inference.

---

> ### Author Response · Authors · 2023-11-13
> **Author Rebuttal**
>
> **Validity of conclusions in our experimental analysis**
>
> We agree with the reviewer and would like to state that we were extremely careful about such issues. We trained the various nuisance models for CATE estimators and metrics using cross-validation with AutoML to be robust to issues like different partitioning of the training and validation set. Further, we have used a large number of random seeds (10 per experiment) to be robust to discrepancy from initialization of model parameters. Finally, since the statistics of ITE estimates, covariate dimension, etc. can change across datasets (Table 4, 5, 6), to determine robust trends we look at a large collection of datasets (78 in total) and train a wide variety of CATE estimators per dataset.
>
> **Data requirements in the proposed framework**
>
> The proposed framework does not in principle need large amounts of data. Our goal was to conduct a comprehensive large-scale evaluation study to make a fair comparison between the various evaluation metrics used in the literature. This helps to create an informative prior for practitioners regarding which evaluation metrics maybe useful for them.
>
> Further, the suggestions of a two-level model selection strategy and causal ensembling do not necessarily need lots of data. It is upto the practitioner to train as many CATE estimators as required for their application and then incorporate the two-level model selection and causal ensembling strategy. Also, with regards to the training of the nuisance models (outcome & propensity models), we are in the typical regime as used by the other works in this area, with the standard train/validation split for the various datasets consider in our work.
>
> **Missing relevant works**
>
> We tried to be as thorough as we could about the prior works and did cover the most relevant ones. We do appreciate the suggestion by the reviewer and are happy to talk about other related works concerning our study.
>
> Regarding the work by Nguyen et al. pointed out by the reviewer, their focus is more on sensitivity analysis, which deals with measuring robustness to unobserved confounders, randomized treatments, etc. While sensitivity analysis is an important practical aspect of causal inference, our goal with model selection is more general since sensitivity analysis alone might not be able to result in a clear distinction between several CATE estimators.

---

> > ### Author Response · Authors · 2023-11-22
> > **Gentle Reminder**
> >
> > Please let us know if your concerns have been resolved with our responses. We are happy to provide any further clarifications if needed.

---

> ### Author Response · Authors · 2023-11-23
> **Gentle Reminder**
>
> We wish to remind the reviewer that there are still a few hours left before the rebuttal period ends. It would be great to get your opinions on our response.

---

### Official Review · Reviewer_5dxk · 2023-11-01

**Soundness:** 3 good
**Presentation:** 3 good
**Contribution:** 3 good
**Rating:** 8
**Confidence:** 3

**Summary:**

This work concerns model selection in conditional average treatment effect (CATE) estimation. Compared with prior works, the authors conduct more comprehensive evaluations involving 78 datasets, 34 metrics and 415 CATE estimators. The authors also propose novel strategies and new metrics for the model selection task. Finally, the author refine the existing evaluation protocols by leveraging advanced generative models and realistic datasets.

**Strengths:**

1. The authors provide a very comprehensive and detailed studies in CATE evaluations. The discovered strategies of model selection can benefit both developers and practitioners. In particular, the observations that T and X scores are dominant and the effectiveness of the proposed 2-level model selection are very interesting. The authors also propose multiple effective metrics and improved evaluation protocols, facilitating development of CATE estimation models.
2. The paper is well-written and easy to follow. Prior works regarding evaluations are thoroughly reviewed and presented.

**Weaknesses:**

1. The deep learning or representation-based CATE methods are excluded. The adopted estimators are numerous indeed, while the underlying machine learning models (Appendix A.2) are rather basic.

**Questions:**

1. The author propose some new metrics (e.g., the "Switch" approach) which seem reasonable but show marginal performance gaps in the evaluation. I am wondering the motivation and if there is other advantage of using these new metrics?
2. Although the typical cross validation method could be non-ideal in CATE estimation in general, I am wondering if reporting those numbers can make this work more complete and highlight the effectiveness of the proposed metrics?

---

> ### Author Response · Authors · 2023-11-13
> **Author Rebuttal**
>
> **Lack of Deep Learning based methods**
>
> We agree with the reviewer that it might be interesting to include deep learning-based CATE methods as well. However, since the scale of the study was already quite large and our primary goal was model selection rather than learning state-of-the-art CATE estimators we decided to focus on basic machine learning models for the nuisance models in the CATE meta-learners.
> Further, the datasets that we work with are mostly tabular datasets, hence basic machine learning techniques with appropriate hyperparameter tuning should have good performance as well. We use state-of-the-art hyperparameter tuning techniques like AutoML to learn well-performing nuisance models.
>
> **Cross-validation results with the ideal metric**
>
> We want to highlight that we compute results with the ideal cross-validation metric (PEHE) with ground-truth ITE estimates, as mentioned at the start of section 5. The ideal PEHE metric is used to normalize the reported scores so they can be made consistent across datasets, and the reported results in Tables 1, 2, and 3 demonstrate the percentage difference in the quality of the chosen estimator by a particular metric compared to the ideal metric. As expected, we can see that none of these metrics are better than the ideal PEHE metric as all the reported scores are greater than zero.
>
> Also, if by the typical cross-validation method, the reviewer meant computing the cross-validation score of the nuisance models (like Mu Score in Schuler et al.), then such an approach is not feasible in our study since we already select over the nuisance models using AutoML and these are shared across the different CATE meta-learners in our experiment. Hence, the cross-validation score of the various nuisance models across CATE meta-learners would be similar.
>
> **Motivation behind new metrics**
>
> Although our results show that we do not gain much with these proposed metrics, Table 3 still shows that scores like TMLE, Qini, and Calibration (DR) are among the set of dominating metrics across all datasets. Hence, they could be considered as a good prior option for a new study and might be even more suitable for some applications that are tied to their specific use case, like learning under extreme propensity regime, etc.
>
> The motivation for including adaptive propensity clipping and targeted learning metrics is the extreme propensity regime, where the metrics that depend on inverse propensity scores (IPW, DR) can become biased. Hence, these proposed metrics can handle the extreme propensity region (very small propensity for the observed treatment) by using estimates from the regression-based learners (S/T Learner) or adding inverse propensity correction (TMLE).
>
> For the Calibration score, the intuition is to match the group ATE across subgroups denoted by the different percentiles of the CATE estimates. We compute the group ATE using DR/TMLE, etc. which is viewed as an unbiased sample of the group ATE, and then we compute the weighted error of the group ATE using the CATE estimator against the unbiased group ATE. For the Qini score, rather than computing the group ATE with the CATE estimator as in Calibration, here we compare the improvement with unbiased group ATE estimates versus the uniform sampling estimate. Intuitively, if there is heterogeneity in the data, then we should expect the policy of uniformly treating an individual to be worse as compared to the policy that assigns treatment to the top-k percentile of the CATE estimates. Hence, the Qini score is qualitatively different than most of the other metrics as it does not directly compare the CATE estimates, rather it evaluates whether the groups most likely to be treated would benefit from the treatment.

---

> > ### Author Response · Authors · 2023-11-22
> > **Gentle reminder**
> >
> > Please let us know if our response clarifies your questions! We are happy to provide any further clarifications if needed.

---

> > > ### Comment · Reviewer_5dxk · 2023-11-22
> > >
> > > I thank the authors for the thorough responses addressing my concerns. I would like to keep my decision.
> > > Regarding the discussion about deep learning based CATE, I agree with the authors that there is a scalability issue, while it may be worth acknowldging it in the paper to make this work more complete.

---

> > > > ### Author Response · Authors · 2023-11-22
> > > >
> > > > Thanks a lot for engaging with our work and for your valuable suggestions! We agree and have updated section 4, page 6 mentioning why we do not consider deep learning-based CATE estimators.

---

### Official Review · Reviewer_i3fD · 2023-11-03

**Soundness:** 3 good
**Presentation:** 3 good
**Contribution:** 3 good
**Rating:** 8
**Confidence:** 3

**Summary:**

The authors proposed several new conditional average treatment effect (CATE) model selection metrics and performed a extensive empirical analysis to judge the performance of the new metrics and the metrics in the existing literature.

**Strengths:**

1. The authors provided a guidance on CATE model selection without access to counterfactual data. This can be useful in practice.
2. The authors included 78 datasets and 34 metrics, and trained 415 CATE estimators for each dataset. The analysis should be comprehensive enough with such large scale.

**Weaknesses:**

The design and results of the experiment don't seem to be convincing to me. First, the authors only used 10 replicates (random seeds) to estimate the mean (standard error) Normalized-PEHE, which is probably not large enough. Second, in Tables 1 and 3, most of the metrics are "dominating" on LaLonde CPS, LaLonde PSID and TWINS while the metrics that are not "dominating" are in the minority. This means most metrics perform similarly on those datasets if I understand correctly, which suggests that the conclusions the authors drew are not well supported.

**Questions:**

1. The third paragraph of section 1: "However, the issue is that these estimators often contradict each other and we would need to perform model selection."
     Is there any literature that demonstrates this issue?
2. Section 3: the proposed model selection metrics seem ad hoc to me. Can you provide some intuition behind these metrics?
3. As I mentioned in the Weakness, 10 random seeds are not enough.
4. As I mentioned in the Weakness, the experiment results are not very informative on LaLonde CPS, LaLonde PSID and TWINS. The authors may want to included the results on other datasets to support their conclusion.

---

> ### Author Response · Authors · 2023-11-13
> **Author Rebuttal**
>
> **Conclusions from experimental results**
>
> We would like to point that we also report results on the multiple ACIC 2016 datasets (75 datasets) in Table 1, 2, and 3 in the main paper, where it does not happen that the set of dominating metrics is bigger than the set of non-dominating metrics. Note that while there is a single column for the ACIC 2016 dataset, it is a collection of 75 datasets, therefore, the trends reported for them are more informative than the Twins and LaLonde datasets combined. Hence, the conclusions that we draw are valid since we do not rely only on the Twin and LaLonde datasets, but we also include the multiple ACIC 2016 datasets.
>
> We agree with the reviewer that we may overestimate the set of dominating metrics by only looking at the Twins and LaLonde datasets. However, to determine the set of globally dominating metrics we look at the metrics that are dominating across all the ACIC 2016 datasets as well as the Twins/LaLonde datasets. It is indeed possible for some metrics to be dominating on a few datasets like Twins and LaLonde, but they are not dominating when evaluated across more datasets like ACIC 2016. Specifically, metrics like Match Score and IPW (and its variants) are dominating on the Twins and LaLonde datasets but they are not dominating on the multiple ACIC 2016 datasets. Hence, for a more useful recommendation to practitioners, we suggest metrics that are dominating across a wide range of datasets. Further, the conclusions we draw about the two-level model selection strategy and causal ensembling improving over the single-level strategy are valid since they hold across all the datasets.
>
> **Number of random seeds used in the study**
>
> As the reviewer stated the scale of the study is quite large, hence 10 random seeds for 78 datasets and 415 CATE estimators per dataset implies a total of $323,700$ CATE estimators trained and evaluated. Therefore, we think that 10 random seeds are enough to determine reliable conclusions given the computational cost of carrying out the study.
>
> However, we will generate results for another set of 10 random seeds (making it a total of 20 random seeds), and report them soon in a few days.
>
> **Intuition behind new metrics**
>
> The motivation for including adaptive propensity clipping and targeted learning metrics is the extreme propensity regime, where the metrics that depend on inverse propensity scores (IPW, DR) can become biased. Hence, these proposed metrics can handle the extreme propensity region (very small propensity for the observed treatment) by using estimates from the regression-based learners (S/T Learner) or adding inverse propensity correction (TMLE).
>
> For the Calibration score, the intuition is to match the group ATE across subgroups denoted by the different percentiles of the CATE estimates. We compute the group ATE using DR/TMLE, etc. which is viewed as an unbiased sample of the group ATE, and then we compute the weighted error of the group ATE using the CATE estimator against the unbiased group ATE. For the Qini score, rather than computing the group ATE with the CATE estimator as in Calibration, here we compare the improvement with unbiased group ATE estimates versus the uniform sampling estimate. Intuitively, if there is heterogeneity in the data, then we should expect the policy of uniformly treating an individual to be worse as compared to the policy that assigns treatment to the top-k percentile of the CATE estimates. Hence, the Qini score is qualitatively different than most of the other metrics as it does not directly compare the CATE estimates, rather it evaluates whether the groups most likely to be treated would benefit from the treatment.
>
> Although our results show that we do not gain much with these proposed metrics, Table 3 still shows that scores like TMLE, Qini, and Calibration (DR) are among the set of dominating metrics across all datasets. Hence, they could be considered as a good prior option for a new study, and might be even more suitable for some applications that are tied to their specific use case, like learning under extreme propensity regime, etc.
>
> **Reference for disagreement between CATE estimators**
>
> Several benchmarking studies (listed below) show this issue where multiple CATE estimators can generate different results, which can be explained since the choice of the nuisance model and their hyperparameters, along with the choice of estimation strategy will give rise to different empirical estimates.
>
> - ACIC Competition: https://arxiv.org/pdf/1707.02641.pdf
> - Meta Learners for CATE estimation: http://proceedings.mlr.press/v130/curth21a/curth21a.pdf
> - Model Selection: https://arxiv.org/pdf/1804.05146.pdf

---

> > ### Author Response · Authors · 2023-11-20
> > **Results with more random seeds**
> >
> > We have updated the draft with results over another set of 10 random seeds, hence a total of 20 random seeds in Tables 10 and 11 in section E (Supplementary). Our conclusions with 20 random seeds are still the same as before with 10 random seeds, hence the conclusions are robust to different random parameter initializations.

---

> ### Author Response · Authors · 2023-11-22
> **Gentle reminder**
>
> Please let us know if your concerns have been resolved, we have addressed the issues you expressed regarding the experimental conclusions with more datasets and random seeds. Our study contains the 75 ACIC 2016 datasets in addition to the Twins and LaLonde datasets, and we now have a total of 20 random seeds per estimator for each dataset, therefore making the conclusions robust. We are happy to address any further questions that you might have.

---

> > ### Comment · Reviewer_i3fD · 2023-11-22
> >
> > Thank you for your response and additional results. I believe my concerns have been addressed so I have updated my ratings accordingly.

---

> ### Author Response · Authors · 2023-11-22
>
> Thanks a lot for engaging with our work and for your suggestions!

---

### Official Review · Reviewer_RQGf · 2023-11-04

**Soundness:** 3 good
**Presentation:** 3 good
**Contribution:** 4 excellent
**Rating:** 8
**Confidence:** 4

**Summary:**

This paper focuses on evaluating CATE estimators.  The author discuss common CATE estimators and propose some additional estimation metrics.  They then describe an evaluation pipeline, using semi-synthetic data from a few sources, getting CATE estimates from 103 estimators, and evaluating the performance of 34 metrics.  They present results across four classes of datasets, evaluating a large number of metrics and comparing single-level model selection to a novel two-level model selection strategy.  The authors come to a few high-level conclusions, including that doubly-robust and TMLE methods tend to perform well and that two-level model selection performs better than single-level.

**Strengths:**

The problem the authors are addressing is important.  The authors correctly point out that much evaluation in the literature is lacking, often performed on synthetic data and comparing to only a small subset of other methods.  The authors overall approach is very principled and comprehensive.  The chosen CATE estimators seem representative, and the authors seem to have made reasonable steps towards evaluating on semi-realistic data.  The narrative flow of the paper is clear and it's overall well-written.  I also really liked the authors' analysis of the empirical results.  The conclusions they drew in Section 5 are interesting and well-described, suggesting real, useful recommendations for the application of CATE estimators.

Based on authors' response, I have updated my score from a 6 to an 8.

**Weaknesses:**

The introduction suffers from a bit of hyperbole.  For example, "This [...] problem is at the heart of EVERY decision making problem", "With the emergence of rich data sets" (implying that 'rich data sets' are new), and "The challenging task is to identify which individuals..." ('The' implies it is the only challenging task - it should 'A challenging task' or even 'An important yet challenging task').

Some terminology clarification could be helpful here.  There are a few terms which, while present in the literature, are not necessarily universally used.  For example, 'nuisance models' and the task being 'model selection' rather than causal modeling/discovery/inference.  Also, the first sentence of Section 2 says that X "represents the controls."  However, 'control' is a pretty overloaded term in causality (generally referring to treatment = 0 in a binary treatment setting), and, especially since you're only considering binary treatment, calling the X's controls just feels unnecessarily confusing.  'Covariates' or 'confounders' would be clearer.  The term 'metric' also feels a bit overloaded and could at least benefit from some clarification of what definition you're using here.  In evaluation literature, I generally expect a 'metric' to be a measure of performance.  So, for example, In Table 1, I would call 'PEHE' the evaluation metric the things under the 'Metric' column 'Estimators'.  Just a sentence or two early on to clarify these definitions would go a long way for increasing clarity.

In Section 2, you define tau(x) as CATE and tau-hat as a CATE estimate.  However, I don't see where you define tau\~, which appears for the first time, I believe, under 'CATE Model Selection Metrics' as a term in the equation.  The next sentence mentions learning tau~, but I don't see anywhere where you describe what it actually represents.  Actually, it looks like you may just be missing the ~ in the sentence before the equation (where it reads: "includes approximating the ground-truth effect (tau(X)) on the validation set").

While I appreciate at some level the inclusion of novel metrics, given that you already are considering quite a lot of metrics, I would have liked more of a discussion as to the motivation for including each of these new ones.  Is there something that's lacking in the existing set of metrics that led you to the new ones?  Or are there aspects of the new metrics that you think could capture something that isn't currently captured?  Performance-wise, none of the IPTW modifications (IPTW Switch T Score, IPTW CAB T Score) appear to do better than just IPTW score, and Qini and Calibration scores rarely do better either.  That doesn't mean that these metrics weren't worth considering, or that the results aren't interesting.  As it is, while IPTW and Targeted Learning at least have a sentence each about why they are being included here (due to their handling of extreme propensities which other metrics don't do), the Calibration Score discussion just says that it "has been studied on RCTs", and Qini just says it comes from the uplift modeling literature.  Without a strong justification for including them, and without them performing particularly well in the results, they just come across as being included for the sake of it, and they contribute little to the paper as a whole.

Given that the two-level model selection strategy is described by the authors as 'novel', I wish a bit more detail were given.  As it is, it says "we first select its hyperparameters using a metric that is designed with similar inductive bias" - what metric?  Does it vary by estimator?  Also, is this a contribution? (it's described as novel but I don't see it listed under 'Contributions' in the introduction)

There are some typos/grammatical issues.  Just looking at the introduction, the first sentence of the 3rd paragraph should say "This has led", not "This has lead", in the 4th paragraph, "which have also been shown to be effect than other metrics" is missing the word 'more', and in the Contribution, the first sentence should not contain a semi-colon (since the phrase that comes after is a dependent clause - just replace with a regular comma).

**Questions:**

In Section 3.1, the IPTW Switch equation shows using the IPW approach if the propensity score is >= epsilon, and the regression learner when it's < epsilon.  Presumably, epsilon is a small number.  However, the sentence after that equation says that you use IPW "if the propensity of the observed treatment for that sample is large."  It looks like the regression learner is being used when the propensity score is small (< epsilon), which means that IPW is being used whenever the propensity score is just 'not small' (>= epsilon), which is very different from it being 'large'.  Am I misunderstanding something here?

In Section 4, under "CATE estimators with well-tuned nuisance models", the 3rd paragraph start with "Further for the CATE estimators with final models (f)" - what does it mean for an estimator to have a 'final model' vs not?

---

> ### Author Response · Authors · 2023-11-13
> **Author Rebuttal**
>
> **Regarding the terminology and writing in the paper**
>
> We thank the reviewer for their detailed feedback regarding the presentation and writing! We agree and would make the writing better with clear definitions and clarifications. We have already updated the paper with these suggestions and the changes can be viewed in the new draft.
>
> Specifically, we refer to X as covariates now as opposed to controls and have added a sentence in section 2 to specify exactly what we mean by metrics for model selection. Further, other suggested changes regarding the writing have been made in the updated draft.
>
>
> **Typo in CATE model selection metrics**
>
> We thank the reviewer for pointing this out, indeed there was a typo in the sentence before the equation and it should have been ($\tilde{\tau}$). The same has been corrected in the updated draft.
>
>
> **IPTW Switch equation and related description**
>
> We agree with the reviewer and have updated the related description. Indeed, the switch score would rely on IPW if the propensity is not very small. We set epsilon to very small values and hence IPW is used when the propensity for the observed treatment is not very small.
>
> &nbsp;
>
> **Final models in CATE estimators**
>
> By final model, we mean the regression model used for predicting CATE in direct meta-learners like DR Learner (eq 15), R Learner (eq 16). For example in DR Learner, we construct the doubly robust potential outcomes ($Y_{0}^{DR}, Y_{1}^{DR}$) and then predict the CATE by learning the regression model ( $f_{\theta} : X \rightarrow  Y_{1}^{DR} - Y_{0}^{DR}$ ) that maps the covariates to doubly robust potential outcomes. This is referred to as the final (regression) model.
>
>  Details regarding the CATE estimators were pushed in section B in the Appendix due to space constraints, but we will try to add a few lines about them in the main text, ideally in section 2. For now, we have referred to the specific equation when mentioning the final model of CATE estimators in Section 4.
>
> **Details regarding the two-level model selection strategy**
>
> The details regarding the two-level model selection strategy are present in Section 4, and the same has been mentioned under the contributions section (bullet 2). But we agree the same could be presented in a better manner, and we have done so in the updated draft.
>
> Indeed, the choice of metric for selecting hyperparameters in the two-level strategy depends on the CATE estimator. Specifically, the hyperparameters for CATE estimators in our study are the final regression models, hence we need to do this for only DR-Learner, R-Learner, X-Learner, and Projected S-Learner, as the others do not have hyperparameters. Now for each of the four CATE estimators mentioned above, we select the hyperparameters using a metric with similar inductive bias, i.e., DR Score for the DR-Learner; R Score for the R-Learner, X Score for the X-Learner, and S Score for the Projected S-Learner.
>
> Essentially, the two-level model selection strategy reduces the population of CATE estimators to different CATE meta-learners, and the choice of metric for reducing the population of a specific CATE meta-learner (e.g. DR-Learner) is designed to maximize the similarity with the inductive bias of the CATE meta-learner, to do reliable hyperparameter optimization.
>
> **Motivation behind new metrics**
>
> Our motivation was to connect widely used ideas from related problems like policy learning, uplift modeling, etc. to have more good candidates for model selection. Although our results show that we do not gain much with these proposed metrics, Table 3 still shows that scores like TMLE, Qini, and Calibration (DR) are among the set of dominating metrics across all datasets. Hence, they could be considered as a good prior option for a new study.
>
> As the reviewer stated, the motivation for including adaptive propensity clipping and targeted learning metrics is the extreme propensity regime. For the Calibration score, the intuition is to match the group ATE across subgroups denoted by the different percentiles of the CATE estimates. We compute the group ATE using DR/TMLE, etc. which is viewed as an unbiased sample of the group ATE. Subsequently, we compute the weighted error of the group ATE using the CATE estimator against the unbiased group ATE.
>
> For the Qini score, rather than computing the group ATE with the CATE estimator as in Calibration, here we compare the improvement with unbiased group ATE estimates versus the uniform sampling estimate. Intuitively, if there is heterogeneity in the data, then we should expect the policy of uniformly treating an individual to be worse as compared to the policy that assigns treatment to the top-k percentile of the CATE estimates. Hence, the Qini score is qualitatively different than most of the other metrics as it does not directly compare the CATE estimates, rather it evaluates whether the groups most likely to be treated would benefit from the treatment.

---

> > ### Comment · Reviewer_RQGf · 2023-11-22
> >
> > Thank you for your detailed response and willingness to incorporate suggestions.  I have updated my score accordingly.

---

> > > ### Author Response · Authors · 2023-11-22
> > >
> > > Thanks a lot for engaging with our work and for the insightful suggestions!

---

### Author Response · Authors · 2023-11-13
**Author Rebuttal**

We thank all the reviewers for their insightful comments and questions. We are happy to see that many reviewers appreciate the significance, analysis, and presentation of our results. For example, **reviewer RQGf** said “The narrative flow of the paper is clear and it's overall well-written. I also really liked the authors' analysis of the empirical results. The conclusions they drew in Section 5 are interesting and well-described, suggesting real, useful recommendations for the application of CATE estimators.” **Reviewer 5dxk** said “The authors provide a very comprehensive and detailed studies in CATE evaluations. The discovered strategies of model selection can benefit both developers and practitioners. In particular, the observations that T and X scores are dominant and the effectiveness of the proposed 2-level model selection are very interesting.” **Reviewer i3fD** said “The authors included 78 datasets and 34 metrics, and trained 415 CATE estimators for each dataset. The analysis should be comprehensive enough with such large scale.” **Reviewer 3TcV** point that “​​This work uses extensive empirical analysis to explore the possibility as well as the best way to do model selection for CATE estimation.”

We address the concerns raised by each reviewer in detail as an individual response to each reviewer. Further, we have updated the draft of our work to clarify some definitions/terminology as suggested by **Reviewer RQGf**. We will again update the draft of the work in a few days to include results with an additional set of 10 random seeds to address the point raised by **Reviewer i3fD**.

&nbsp;

Since the concern of motivation behind new metrics was shared by multiple reviewers [**RQGf, i3fD, 5dxk**], we respond to it below.

**The motivation behind the proposed metrics**

The primary reason for including these new metrics was to have a more comprehensive evaluation, not necessarily to beat the prior metrics. The ideas of switching, blending, Qini, etc. have been well-studied and extensively used in the policy learning literature, but their application in the context of CATE estimation/ model selection has not been shown. Similarly, targeted learning has been used for learning ATE/CATE estimators but it remains unexplored as a model selection metric. Hence, our motivation was to connect widely used ideas from related problems like policy learning, uplift modeling, etc. to have more good candidates for model selection.

The motivation for including adaptive propensity clipping and targeted learning metrics is the extreme propensity regime, where the metrics that depend on inverse propensity scores (IPW, DR) can become biased. Hence, these proposed metrics can handle the extreme propensity region (very small propensity for the observed treatment) by using estimates from the regression-based learners (S/T Learner) or adding inverse propensity correction (TMLE).

For the Calibration score, the intuition is to match the group ATE across subgroups denoted by the different percentiles of the CATE estimates. We compute the group ATE using DR/TMLE, etc. which is viewed as an unbiased sample of the group ATE. Subsequently, we compute the weighted error of the group ATE using the CATE estimator against the unbiased group ATE. For the Qini score, rather than computing the group ATE with the CATE estimator as in Calibration, here we compare the improvement with unbiased group ATE estimates versus the uniform sampling estimate. Intuitively, if there is heterogeneity in the data, then we should expect the policy of uniformly treating an individual to be worse as compared to the policy that assigns treatment to the top-k percentile of the CATE estimates. Hence, the Qini score is qualitatively different than most of the other metrics as it does not directly compare the CATE estimates, rather it evaluates whether the groups most likely to be treated would benefit from the treatment.

Although our results show that we do not gain much with these proposed metrics, Table 3 still shows that scores like TMLE, Qini, and Calibration (DR) are among the set of dominating metrics across all datasets. Hence, they could be considered as a good prior option for a new study, and might be even more suitable for some applications that are tied to their specific use case, like learning under extreme propensity regime, etc.

---

> ### Author Response · Authors · 2023-11-20
> **Results with more random seeds**
>
> To address the point raised by **reviewer i3fD**, we have updated the draft with results over another set of 10 random seeds, hence a total of 20 random seeds in Tables 10 and 11 in section E (Supplementary). Our conclusions with 20 random seeds are still the same as before with 10 random seeds, hence the conclusions are robust to different random parameter initializations.

---

### Author Response · Authors · 2023-11-20
**Gentle reminder to reviewers**

We wish to remind the reviewers that there is still time left before the discussion period ends. Please look at our responses and kindly consider increasing your score if your concerns are addressed. We are happy to provide further clarifications!

---

### Meta-Review · Area_Chair_EJKp · 2023-12-07

**Metareview:**

Through a number of experimental analyses, this paper evaluates various metrics of CATE estimators and proposes several new metrics. The paper is useful from both a academic and practical perspectives, and is also well-written, and would be of high value to be shared within the community.

**Justification For Why Not Higher Score:**

This paper may not be of much interest to theorists.

**Justification For Why Not Lower Score:**

Findings based on relatively exhaustive experimental results will be of interest to many.

---

### Decision · Program_Chairs · 2024-01-16

Accept (spotlight)